# Acceleration through Optimistic No-Regret Dynamics

**Jun-Kun Wang**
College of Computing
Georgia Institute of Technology
Atlanta, GA 30313
jimwang@gatech.edu

**Jacob Abernethy**
College of Computing
Georgia Institute of Technology
Atlanta, GA 30313
prof@gatech.edu

## Abstract

We consider the problem of minimizing a smooth convex function by reducing the optimization to computing the Nash equilibrium of a particular zero-sum convex-concave game. Zero-sum games can be solved using online learning dynamics, where a classical technique involves simulating two no-regret algorithms that play against each other and, after $T$ rounds, the average iterate is guaranteed to solve the original optimization problem with error decaying as $O(\log T/T)$. In this paper we show that the technique can be enhanced to a rate of $O(1/T^2)$ by extending recent work [22, 25] that leverages *optimistic learning* to speed up equilibrium computation. The resulting optimization algorithm derived from this analysis coincides *exactly* with the well-known NESTEROVACCELERATION [16] method, and indeed the same story allows us to recover several variants of the Nesterov's algorithm via small tweaks. We are also able to establish the accelerated linear rate for a function which is both strongly-convex and smooth. This methodology unifies a number of different iterative optimization methods: we show that the HEAVYBALL algorithm is precisely the non-optimistic variant of NESTEROVACCELERATION, and recent prior work already established a similar perspective on FRANKWOLFE [2, 1].

## 1 Introduction

One of the most successful and broadly useful tools recently developed within the machine learning literature is the *no-regret framework*, and in particular *online convex optimization* (OCO) [28]. In the standard OCO setup, a learner is presented with a sequence of (convex) loss functions $\ell_1(\cdot), \ell_2(\cdot), \ldots$, and must make a sequence of decisions $x_1, x_2, \ldots$ from some set $\mathcal{K}$ in an online fashion, and observes $\ell_t$ after only having committed to $x_t$. Assuming the sequence $\{\ell_t\}$ is chosen by an adversary, the learner aims is to minimize the *average regret* $\bar{R}_T := \frac{1}{T}\left(\sum_{t=1}^{T} \ell_t(x_t) - \min_{x \in \mathcal{K}} \sum_{t=1}^{T} \ell_t(x)\right)$ against any such loss functions. Many simple algorithms have been developed for OCO problems—including MIRRORDESCENT, FOLLOWTHEREGULARIZEDLEADER, FOLLOWTHEPERTURBEDLEADER, etc.—and these algorithms exhibit regret guarantees that are strong even against adversarial opponents. Under very weak conditions one can achieve a regret rate of $\bar{R}_T = O(1/\sqrt{T})$, or even $\bar{R}_T = O(\log T/T)$ with required curvature on $\ell_t$.

One can apply online learning tools to several problems, but perhaps the simplest is to find the approximate minimum of a convex function $\arg\min_{x \in \mathcal{K}} f(x)$. With a simple reduction we set $\ell_t = f$, and it is easy to show that, via Jensen's inequality, the average iterate $\bar{x}_T := \frac{x_1 + \ldots + x_T}{T}$ satisfies

$$f(\bar{x}_T) \le \tfrac{1}{T}\sum_{t=1}^{T} f(x_t) = \tfrac{1}{T}\sum_{t=1}^{T} \ell_t(x_t) \le \min_{x \in \mathcal{K}} \tfrac{1}{T}\sum_{t=1}^{T} \ell_t(x) + \bar{R}_T = \min_{x \in \mathcal{K}} f(x) + \bar{R}_T$$

hence $\bar{R}_T$ upper bounds the approximation error. But this reduction, while simple and natural, is quite limited. For example, we know that when $f(\cdot)$ is *smooth*, more sophisticated algorithms such

as FRANKWOLFE and HEAVYBALL achieve convergence rates of $O(1/T)$, whereas the now-famous NESTEROVACCELERATION algorithm achieves a rate of $O(1/T^2)$. The fast rate shown by Nesterov was quite surprising at the time, and many researchers to this day find the result quite puzzling. There has been a great deal of work aimed at providing a more natural explanation of acceleration, with a more intuitive convergence proof [27, 4, 10]. This is indeed one of the main topics of the present work, and we will soon return to this discussion.

Another application of the no-regret framework is the solution of so-called saddle-point problems, which are equivalently referred to as Nash equilibria for zero-sum games. Given a function $g(x, y)$ which is convex in $x$ and concave in $y$ (often called a *payoff function*), define $V^* = \inf_{x \in \mathcal{K}} \sup_y g(x, y)$. An $\epsilon$-*equilibrium* of $g(\cdot, \cdot)$ is a pair $\hat{x}, \hat{y}$ such that such that

$$V^* - \epsilon \leq \inf_{x \in \mathcal{K}} g(x, \hat{y}) \leq V^* \leq \sup_y g(\hat{x}, y) \leq V^* + \epsilon. \tag{1}$$

One can find an approximate saddle point of the game with the following setup: implement a no-regret learning algorithm for both the $x$ and $y$ players simultaneously, after observing the actions $\{x_t, y_t\}_{t=1...T}$ return the time-averaged iterates $(\hat{x}, \hat{y}) = \left(\frac{x_1 + ... + x_T}{T}, \frac{y_1 + ... + y_T}{T}\right)$. A simple proof shows that $(\hat{x}, \hat{y})$ is an approximate equilibrium, with approximation bounded by the average regret of both players (see Theorem 1). In the case where the function $g(\cdot, \cdot)$ is biaffine, the no-regret reduction guarantees a rate of $O(1/\sqrt{T})$, and it was assumed by many researchers this was the fastest possible using this framework. But one of the most surprising online learning results to emerge in recent years established that no-regret dynamics can obtain an even faster rate of $O(1/T)$. Relying on tools developed by [8], this fact was first proved by [21] and extended by [25]. The new ingredient in this recipe is the use of *optimistic* learning algorithms, where the learner seeks to benefit from the predictability of slowly-changing inputs $\{\ell_t\}$.

We will consider solving the classical convex optimization problem $\min_x f(x)$, for smooth functions $f$, by instead solving an associated saddle-point problem which we call the *Fenchel Game*. Specifically, we consider that the payoff function $g$ of the game to be

$$g(x, y) = \langle x, y \rangle - f^*(y). \tag{2}$$

where $f^*(\cdot)$ is the *fenchel conjugate* of $f(\cdot)$. This is an appropriate choice of payoff function since, $V^* = \min_x f(x)$ and $\sup_y g(\hat{x}, y) = \sup_y \langle \hat{x}, y \rangle - f^*(y) = f(\hat{x})$. Therefore, by the definition of an $\epsilon$-equilibrium, we have that

**Lemma 1.** *If $(\hat{x}, \hat{y})$ is an $\epsilon$-equilibrium of the Fenchel Game* (2)*, then $f(\hat{x}) - \min_x f(x) \leq \epsilon$.*

One can imagine computing the equilibrium of the Fenchel game using no-regret dynamics, and indeed this was the result of recent work [2] establishing the FRANKWOLFE algorithm as precisely an instance of two competing learning algorithms.

In the present work we will take this approach even further.

1. We show that, by considering a notion of *weighted regret*, we can compute equilibria in the Fenchel game at a rate of $O(1/T^2)$ using no-regret dynamics where the only required condition is that $f$ is smooth. This improves upon recent work [1] on a faster FRANKWOLFE method, which required strong convexity of $f$ (see Appendix J).
2. We show that the secret sauce for obtaining the fast rate is precisely the use of an optimistic no-regret algorithm, OPTIMISTICFTL [1], combined with appropriate weighting scheme.
3. We show that, when viewed simply as an optimization algorithm, this method is *identically* the original NESTEROVACCELERATION method. In addition, we recover several variants of NESTEROVACCELERATION (see [15, 17, 19]) using small tweaks of the framework.
4. We show that if one simply plays FOLLOWTHELEADER without optimism, the resulting algorithm is precisely the HEAVYBALL. The latter is known to achieve a suboptimal rate in general, and our analysis sheds light on this difference.
5. Under the additional assumption that function $f(\cdot)$ is strongly convex, we show that an accelerated linear rate can also be obtained from the game framework.
6. Finally, we show that the same equilibrium framework can also be extended to composite optimization and lead to a variant of Accelerated Proximal Method.

**Related works:** In recent years, there are growing interest in giving new interpretations of Nesterov's accelerated algorithms. For example, [26] gives a unified analysis for some Nesterov's accelerated

algorithms [17, 18, 19], using the standard techniques and analysis in optimization literature. [13] connects the design of accelerated algorithms with dynamical systems and control theory. [7] gives a geometric interpretation of the Nesterov's method for unconstrained optimization, inspired by the ellipsoid method. [10] studies the Nesterov's methods and the HEAVYBALL method for quadratic non-strongly convex problems by analyzing the eigen-values of some linear dynamical systems. [4] proposes a variant of accelerated algorithms by mixing the updates of gradient descent and mirror descent and showing the updates are complementary. [24, 27] connect the acceleration algorithms with differential equations. In recent years there has emerged a lot of work where learning problems are treated as repeated games [14, 3], and many researchers have been studying the relationship between game dynamics and provable convergence rates [5, 11, 9].

We would like to acknowledge George Lan for his excellent notes titled "Lectures on Optimization for Machine Learning" (unpublished). In parallel to the development of the results in this paper, we discovered that Lan had observed a similar connection between NESTEROVACCELERATION and repeated game playing (Chapter 3.4). A game interpretation was given by George Lan and Yi Zhou in Section 2.2 of [12].

## 2 Preliminaries

**Convex functions and conjugates.** A function $f$ on $\mathbb{R}^d$ is $L$-smooth w.r.t. a norm $\|\cdot\|$ if $f$ is everywhere differentiable and it has lipschitz continuous gradient $\|\nabla f(u) - \nabla f(v)\|_* \leq L\|u - v\|$, where $\|\cdot\|_*$ denotes the dual norm. Throughout the paper, our goal will be to solve the problem of minimizing an $L$-smooth function $f(\cdot)$ over a convex set $\mathcal{K}$. We also assume that the optimal solution of $x^* := \operatorname{argmin}_{x \in \mathcal{K}} f(x)$ has finite norm. For any convex function $f$, its Fenchel conjugate is $f^*(y) := \sup_{x \in \operatorname{dom}(f)} \langle x, y \rangle - f(x)$. If a function $f$ is convex, then its conjugate $f^*$ is also convex. Furthermore, when the function $f(\cdot)$ is strictly convex, we have that $\nabla f(x) = \operatorname{argmax}_{y} \langle x, y \rangle - f^*(y)$.

Suppose we are given a differentiable function $\phi(\cdot)$, then the Bregman divergence $V_c(x)$ with respect to $\phi(\cdot)$ at a point $c$ is defined as $V_c(x) := \phi(x) - \langle \nabla \phi(c), x - c \rangle - \phi(c)$. Let $\|\cdot\|$ be any norm on $\mathbb{R}^d$. When we have that $V_c(x) \geq \frac{\sigma}{2}\|c - x\|^2$ for any $x, c \in \operatorname{dom}(\phi)$, we say that $\phi(\cdot)$ is a $\sigma$-*strongly convex function* with respect to $\|\cdot\|$. Throughout the paper we assume that $\phi(\cdot)$ is 1-strongly convex.

**No-regret zero-sum game dynamics.** Let us now consider the process of solving a zero-sum game via repeatedly play by a pair of online learning strategies. The sequential procedure is described in Algorithm 1.

---

**Algorithm 1** Computing equilibrium using no-regret algorithms

1: Input: sequence $\alpha_1, \ldots, \alpha_T > 0$
2: **for** $t = 1, 2, \ldots, T$ **do**
3:     $y$-player selects $y_t \in \mathcal{Y} = \mathbb{R}^d$ by OAlg$^y$.
4:     $x$-player selects $x_t \in \mathcal{X}$ by OAlg$^x$, possibly with knowledge of $y_t$.
5:     $y$-player suffers loss $\ell_t(y_t)$ with weight $\alpha_t$, where $\ell_t(\cdot) = -g(x_t, \cdot)$.
6:     $x$-player suffers loss $h_t(x_t)$ with weight $\alpha_t$, where $h_t(\cdot) = g(\cdot, y_t)$.
7: **end for**
8: Output $(\bar{x}_T, \bar{y}_T) := \left( \frac{\sum_{s=1}^T \alpha_s x_s}{A_T}, \frac{\sum_{s=1}^T \alpha_s y_s}{A_T} \right)$.

---

In this paper, we consider Fenchel game with weighted losses depicted in Algorithm 1, following the same setup as [1]. In this game, the $y$-player plays before the $x$-player plays and the $x$-player sees what the $y$-player plays before choosing its action. The $y$-player receives loss functions $\alpha_t \ell_t(\cdot)$ in round $t$, in which $\ell_t(y) := f^*(y) - \langle x_t, y \rangle$, while the x-player see its loss functions $\alpha_t h_t(\cdot)$ in round $t$, in which $h_t(x) := \langle x, y_t \rangle - f^*(y_t)$. Consequently, we can define the *weighted regret* of the $x$ and $y$ players as

$$\boldsymbol{\alpha}\text{-}\mathrm{REG}^y \quad := \quad \sum_{t=1}^T \alpha_t \ell_t(y_t) - \min_y \sum_{t=1}^T \alpha_t \ell_t(y) \qquad (3)$$

$$\boldsymbol{\alpha}\text{-}\mathrm{REG}^x \quad := \quad \sum_{t=1}^T \alpha_t h_t(x_t) - \sum_{t=1}^T \alpha_t h_t(x^*) \qquad (4)$$

Notice that the $x$-player's regret is computed relative to $x^*$ the minimizer of $f(\cdot)$, rather than the minimizer of $\sum_{t=1}^{T} \alpha_t h_t(\cdot)$. Although slightly non-standard, this allows us to handle the unconstrained setting while Theorem 1 still holds as desired.

At times when we want to refer to the regret on another sequence $y_1', \ldots, y_T'$ we may refer to this as $\boldsymbol{\alpha}\text{-REG}(y_1', \ldots, y_T')$. We also denote $A_t$ as the cumulative sum of the weights $A_t := \sum_{s=1}^{t} \alpha_s$ and the weighted average regret $\overline{\boldsymbol{\alpha}\text{-REG}} := \frac{\boldsymbol{\alpha}\text{-REG}}{A_T}$. Finally, for offline constrained optimization (i.e. $\min_{x \in \mathcal{K}} f(x)$), we let the decision space of the benchmark/comparator in the weighted regret definition to be $\mathcal{X} = \mathcal{K}$; for offline unconstrained optimization, we let the decision space of the benchmark/comparator to be a norm ball that contains the optimum solution of the offline problem (i.e. contains $\arg\min_{x \in \mathbb{R}^n} f(x)$), which means that $\mathcal{X}$ of the comparator is a norm ball. We let $\mathcal{Y} = \mathbb{R}^d$ be unconstrained.

**Theorem 1.** *[1] Assume a $T$-length sequence $\boldsymbol{\alpha}$ are given. Suppose in Algorithm 1 the online learning algorithms $\text{OAlg}^x$ and $\text{OAlg}^y$ have the $\boldsymbol{\alpha}$-weighted average regret $\overline{\boldsymbol{\alpha}\text{-REG}}^x$ and $\overline{\boldsymbol{\alpha}\text{-REG}}^y$ respectively. Then the output $(\bar{x}_T, \bar{y}_T)$ is an $\epsilon$-equilibrium for $g(\cdot, \cdot)$, with $\epsilon = \overline{\boldsymbol{\alpha}\text{-REG}}^x + \overline{\boldsymbol{\alpha}\text{-REG}}^y$.*

# 3 An Accelerated Solution to the Fenchel Game via Optimism

We are going to analyze more closely the use of Algorithm 1, with the help of Theorem 1, to establish a fast method to compute an approximate equilibrium of the Fenchel Game. In particular, we will establish an approximation factor of $O(1/T^2)$ after $T$ iterations, and we recall that this leads to a $O(1/T^2)$ algorithm for our primary goal of solving $\min_{x \in \mathcal{K}} f(x)$.

## 3.1 Analysis of the weighted regret of the y-player (i.e. the gradient player)

A very natural online learning algorithm is FOLLOWTHELEADER, which always plays the point with the lowest (weighted) historical loss

$$\text{FOLLOWTHELEADER} \qquad \hat{y}_t \quad := \quad \text{argmin}_y \left\{ \sum_{s=1}^{t-1} \alpha_s \ell_s(y) \right\}.$$

FOLLOWTHELEADER is known to not perform well against arbitrary loss functions, but for strongly convex $\ell_t(\cdot)$ one can prove an $O(\log T / T)$ regret bound in the unweighted case. For the time being, we shall focus on a slightly different algorithm that utilizes "optimism" in selecting the next action:

$$\text{OPTIMISTICFTL} \qquad \widetilde{y}_t \quad := \quad \text{argmin}_y \left\{ \alpha_t \ell_{-1}(y) + \sum_{s=1}^{t-1} \alpha_s \ell_s(y) \right\}.$$

This procedure can be viewed as an optimistic variant of FOLLOWTHELEADER since the algorithm is effectively making a bet that, while $\ell_t(\cdot)$ has not yet been observed, it is likely to be quite similar to $\ell_{t-1}$. Within the online learning community, the origins of this trick go back to [8], although their algorithm was described in terms of a 2-step descent method. This was later expanded by [21] who coined the term *optimistic mirror descent* (OMD), and who showed that the proposed procedure can accelerate zero-sum game dynamics when both players utilize OMD. OPTIMISTICFTL, defined as a "batch" procedure, was first presented in [1] and many of the tools of the present paper follow directly from that work.

For convenience, we'll define $\delta_t(y) := \alpha_t(\ell_t(y) - \ell_{t-1}(y))$. Intuitively, the regret will be small if the functions $\delta_t$ are not too big. This is formalized in the following lemma.

**Lemma 2.** *For an arbitrary sequence $\{\alpha_t, \ell_t\}_{t=1 \ldots T}$, the regret of OPTIMISTICFTL satisfies $\boldsymbol{\alpha}\text{-REG}^y(\widetilde{y}_1, \ldots, \widetilde{y}_T) \leq \sum_{t=1}^{T} \delta_t(\widetilde{y}_t) - \delta_t(\hat{y}_{t+1})$.*

*Proof.* Let $L_t(y) := \sum_{s=1}^{t} \alpha_s \ell_s(y)$ and also $\tilde{L}_t(y) := \alpha_t \ell_{t-1}(y) + \sum_{s=1}^{t-1} \alpha_s \ell_s(y)$.

$$
\begin{aligned}
\boldsymbol{\alpha}\text{-REG}(\widetilde{y}_{1:T}) \quad &:= \quad \sum_{t=1}^{T} \alpha_t \ell_t(\widetilde{y}_t) - L_T(\hat{y}_{T+1}) = \sum_{t=1}^{T} \alpha_t \ell_t(\widetilde{y}_t) - \tilde{L}_T(\hat{y}_{T+1}) - \delta_T(\hat{y}_{T+1}) \\
&\leq \quad \sum_{t=1}^{T} \alpha_t \ell_t(\widetilde{y}_t) - \tilde{L}_T(\widetilde{y}_T) - \delta_T(\hat{y}_{T+1}) \\
&= \quad \sum_{t=1}^{T-1} \alpha_t \ell_t(\widetilde{y}_t) - L_{T-1}(\widetilde{y}_T) + \delta_T(\widetilde{y}_T) - \delta_T(\hat{y}_{T+1}) \\
&\leq \quad \sum_{t=1}^{T-1} \alpha_t \ell_t(\widetilde{y}_t) - L_{T-1}(\hat{y}_T) + \delta_T(\widetilde{y}_T) - \delta_T(\hat{y}_{T+1}) \\
&= \quad \boldsymbol{\alpha}\text{-REG}(\widetilde{y}_{1:T-1}) + \delta_T(\widetilde{y}_T) - \delta_T(\hat{y}_{T+1}).
\end{aligned}
$$

The bound follows by induction on $T$. □

The result from Lemma 2 is generic, and would hold for any online learning problem. But for the Fenchel game, we have a very specific sequence of loss functions, $\ell_t(y) := -g(x_t, y) = f^*(y) - \langle x_t, y \rangle$. With this in mind, let us further analyze the regret of the $y$ player.

For the time being, let us assume that the sequence of $x_t$'s is arbitrary. We define

$$\bar{x}_t := \frac{1}{A_t} \sum_{s=1}^{t} \alpha_s x_s \qquad \text{and} \qquad \widetilde{x}_t := \frac{1}{A_t}(\alpha_t x_{t-1} + \sum_{s=1}^{t-1} \alpha_s x_s).$$

It is critical that we have two parallel sequences of iterate averages for the $x$-player. Our final algorithm will output $\bar{x}_T$, whereas the Fenchel game dynamics will involve computing $\nabla f$ at the *reweighted averages* $\widetilde{x}_t$ for each $t = 1, \dots, T$.

To prove the key regret bound for the $y$-player, we first need to state some simple technical facts.

$$\hat{y}_{t+1} = \underset{y}{\operatorname{argmin}} \sum_{s=1}^{t} \alpha_s \left( f^*(y) - \langle x_s, y \rangle \right) = \underset{y}{\operatorname{argmax}} \langle \bar{x}_t, y \rangle - f^*(y) = \nabla f(\bar{x}_t) \quad (5)$$

$$\widetilde{y}_t = \nabla f(\widetilde{x}_t) \qquad \text{(following same reasoning as above)}, \quad (6)$$

$$\widetilde{x}_t - \bar{x}_t = \frac{\alpha_t}{A_t}(x_{t-1} - x_t). \quad (7)$$

Equations 5 and 6 follow from elementary properties of Fenchel conjugation and the Legendre transform [23]. Equation 7 follows from a simple algebraic calculation.

**Lemma 3.** *Suppose $f(\cdot)$ is a convex function that is L-smooth with respect to the the norm $\| \cdot \|$ with dual norm $\| \cdot \|_*$. Let $x_1, \dots, x_T$ be an arbitrary sequence of points. Then, we have*

$$\boldsymbol{\alpha}\text{-}\mathrm{REG}^y(\widetilde{y}_1, \dots, \widetilde{y}_T) \leq L \sum_{t=1}^{T} \frac{\alpha_t^2}{A_t} \|x_{t-1} - x_t\|^2. \quad (8)$$

*Proof.* Following Lemma 2, and noting that here we have $\delta_t(y) = \alpha_t \langle x_{t-1} - x_t, y \rangle$, we have

$$
\begin{aligned}
\sum_{t=1}^{T} \alpha_t \ell_t(\widetilde{y}_t) - \alpha_t \ell_t(y^*) &\leq \sum_{t=1}^{T} \delta_t(\widetilde{y}_t) - \delta_t(\hat{y}_{t+1}) = \sum_{t=1}^{T} \alpha_t \langle x_{t-1} - x_t, \widetilde{y}_t - \hat{y}_{t+1} \rangle \\
\text{(Eqns. 5, 6)} \quad &= \sum_{t=1}^{T} \alpha_t \langle x_{t-1} - x_t, \nabla f(\widetilde{x}_t) - \nabla f(\bar{x}_t) \rangle \\
\text{(Hölder's Ineq.)} \quad &\leq \sum_{t=1}^{T} \alpha_t \|x_{t-1} - x_t\| \|\nabla f(\widetilde{x}_t) - \nabla f(\bar{x}_t)\|_* \\
\text{(L-smoothness of } f) \quad &\leq L \sum_{t=1}^{T} \alpha_t \|x_{t-1} - x_t\| \|\widetilde{x}_t - \bar{x}_t\| \\
\text{(Eqn. 7)} \quad &= L \sum_{t=1}^{T} \frac{\alpha_t^2}{A_t} \|x_{t-1} - x_t\| \|x_{t-1} - x_t\|
\end{aligned}
$$

as desired. □

We notice that a similar bound is given in [1] for the gradient player using OPTIMISTICFTL, yet the above result is a stict improvement as the previous work relied on the additional assumption that $f(\cdot)$ is strongly convex. The above lemma depends only on the fact that $f$ has lipschitz gradients.

### 3.2 Analysis of the weighted regret of the x-player

In the present section we are going to consider that the $x$-player uses MIRRORDESCENT for updating its action, which is defined as follows.

$$x_t := \operatorname{argmin}_{x \in \mathcal{K}} \alpha_t h_t(x) + \frac{1}{\gamma_t} V_{x_{t-1}}(x) = \operatorname{argmin}_{x \in \mathcal{K}} \gamma_t \langle x, \alpha_t y_t \rangle + V_{x_{t-1}}(x), \quad (9)$$

where we recall that the Bregman divergence $V_x(\cdot)$ is with respect to a 1-strongly convex regularization $\phi$. Also, we note that the $x$-player has an advantage in these game dynamics, since $x_t$ is chosen *with knowledge of* $y_t$ and hence has knowledge of the incoming loss $h_t(\cdot)$.

**Lemma 4.** *Let the sequence of $x_t$'s be chosen according to MIRRORDESCENT. Assume that the Bregman Divergence is uniformly bounded on $\mathcal{K}$, so that $D = \sup_{t=1,\dots,T} V_{x_t}(x^*)$, where $x^*$ denotes the minimizer of $f(\cdot)$. Assume that the sequence $\{\gamma_t\}_{t=1,2,\dots}$ is non-increasing. Then we have $\boldsymbol{\alpha}\text{-}\mathrm{REG}^x \leq \frac{D}{\gamma_T} - \sum_{t=1}^{T} \frac{1}{2\gamma_t} \|x_{t-1} - x_t\|^2$.*

The proof of this lemma is quite standard, and we postpone it to Appendix A. We also note that the benchmark $x^*$ is always within a finite norm ball by assumption. We given an alternative to this lemma in the appendix, when $\gamma_t = \gamma$ is fixed, in which case we can instead use the more natural constant $D = V_{x_1}(x^*)$.

### 3.3 Convergence Rate of the Fenchel Game

**Theorem 2.** *Let us consider the output $(\bar{x}_T, \bar{y}_T)$ of Algorithm 1 under the following conditions: (a) the sequence $\{\alpha_t\}$ is positive but otherwise arbitrary (b) OAlg$^y$ is chosen OPTIMISTICFTL, (c) OAlg$^x$ is MIRRORDESCENT with any non-increasing positive sequence $\{\gamma_t\}$, and (d) we have a bound $V_{x_t}(x^*) \leq D$ for all $t$. Then the point $\bar{x}_T$ satisfies*

$$f(\bar{x}_T) - \min_{x \in \mathcal{X}} f(x) \leq \frac{1}{A_T} \left( \frac{D}{\gamma_T} + \sum_{t=1}^{T} \left( \frac{\alpha_t^2}{A_t} L - \frac{1}{2\gamma_t} \right) \|x_{t-1} - x_t\|^2 \right). \tag{10}$$

*Proof.* We have already done the hard work to prove this theorem. Lemma 1 tells us we can bound the error of $\bar{x}_T$ by the $\epsilon$ error of the approximate equilibrium $(\bar{x}_T, \bar{y}_T)$. Theorem 1 tells us that the pair $(\bar{x}_T, \bar{y}_T)$ derived from Algorithm 1 is controlled by the sum of averaged regrets of both players, $\frac{1}{A_T}(\boldsymbol{\alpha}\text{-}\text{REG}^x + \boldsymbol{\alpha}\text{-}\text{REG}^y)$. But we now have control over both of these two regret quantities, from Lemmas 3 and 4. The right hand side of (10) is the sum of these bounds. $\square$

Theorem 2 is somewhat opaque without a specifying the sequence $\{\alpha_t\}$. But what we now show is that the summation term *vanishes* when we can guarantee that $\frac{\alpha_t^2}{A_t}$ remains constant! This is where we obtain the following fast rate.

**Corollary 1.** *Following Theorem 2 with $\alpha_t = t$ and for any non-increasing sequence $\gamma_t$ satisfying $\frac{1}{CL} \leq \gamma_t \leq \frac{1}{4L}$ for some constant $C > 4$, we have $f(\bar{x}_T) - \min_{x \in \mathcal{X}} f(x) \leq \frac{2CLD}{T^2}$.*

*Proof.* Observing $A_t := \frac{t(t+1)}{2}$, the choice of $\{\alpha_t, \gamma_t\}$ implies $\frac{D}{\gamma_t} \leq cDL$ and $\frac{L\alpha_t^2}{A_t} = \frac{2Lt^2}{t(t+1)} \leq 2L \leq \frac{1}{2\gamma_t}$, which ensures that the summation term in (10) is negative. The rest is simple algebra. $\square$

A straightforward choice for the learning rate $\gamma_t$ is simple the constant sequence $\gamma_t = \frac{1}{4L}$. The corollary is stated with a changing $\gamma_t$ in order to bring out a connection to the classical NESTEROVAC-CELERATION in the following section.

**Remark:** It is worth dwelling on exactly how we obtained the above result. A less refined analysis of the MIRRORDESCENT algorithm would have simply ignored the negative summation term in Lemma 4, and simply upper bounded this by 0. But the negative terms $\|x_t - x_{t-1}\|^2$ in this sum happen to correspond *exactly* to the positive terms one obtains in the regret bound for the $y$-player, but this is true *only as a result of* using the OPTIMISTICFTL algorithm. To obtain a cancellation of these terms, we need a $\gamma_t$ which is roughly constant, and hence we need to ensure that $\frac{\alpha_t^2}{A_t} = O(1)$. The final bound, of course, is determined by the inverse quantity $\frac{1}{A_T}$, and a quick inspection reveals that the best choice of $\alpha_t = \theta(t)$. This is not the only choice that could work, and we conjecture that there are scenarios in which better bounds are achievable for different $\alpha_t$ tuning. We show in Section 4.3 that a *linear rate* is achievable when $f(\cdot)$ is also strongly convex, and there we tune $\alpha_t$ to grow exponentially in $t$ rather than linearly.

## 4 Nesterov's methods are instances of our accelerated solution to the game

Starting from 1983, Nesterov has proposed three accelerated methods for smooth convex problems (i.e. [16, 15, 17, 19]. In this section, we show that our accelerated algorithm to the *Fenchel game* can generate all his methods with some simple tweaks.

## 4.1 Recovering Nesterov's (1983) method for unconstrained smooth convex problems [16, 15]

In this subsection, we assume that the x-player's action space is unconstrained. That is, $\mathcal{K} = \mathbb{R}^n$. Consider the following algorithm.

---
**Algorithm 2** A variant of our accelerated algorithm.

---
1: In the weighted loss setting of Algorithm 1:
2:     $y$-player uses OPTIMISITCFTL as OAlg$^y$: $y_t = \nabla f(\tilde{x}_t)$.
3:     $x$-player uses ONLINEGRADIENTDESCENT as OAlg$^x$:
4:       $x_t = x_{t-1} - \gamma_t \alpha_t \nabla h_t(x) = x_{t-1} - \gamma_t \alpha_t y_t = x_{t-1} - \gamma_t \alpha_t \nabla f(\tilde{x}_t)$.

---

**Theorem 3.** *Let $\alpha_t = t$. Assume $\mathcal{K} = \mathbb{R}^n$. Algorithm 2 is actually the case the x-player uses* MIRRORDESCENT. *Therefore, $\bar{x}_T$ is an $O(\frac{1}{T^2})$-approximate optimal solution of $\min_x f(x)$ by Theorem 2 and Corollary 1.*

*Proof.* For the unconstrained case, we can let the distance generating function of the Bregman divergence to be the squared of L2 norm, i.e. $\phi(x) := \frac{1}{2}\|x\|_2^2$. Then, the update becomes $x_t = \operatorname{argmin}_x \gamma_t \langle x, \alpha_t y_t \rangle + V_{x_{t-1}}(x) = \operatorname{argmin}_x \gamma_t \langle x, \alpha_t y_t \rangle + \frac{1}{2}\|x\|_2^2 - \langle x_{t-1}, x - x_{t-1} \rangle - \frac{1}{2}\|x_{t-1}\|_2^2$. Differentiating the objective w.r.t $x$ and setting it to zero, one will get $x_t = x_{t-1} - \gamma_t \alpha_t y_t$. $\square$

Having shown that Algorithm 2 is actually our accelerated algorithm to the *Fenchel game*. We are going to show that Algorithm 2 has a direct correspondence with Nesterov's first acceleration method (Algorithm 3) [16, 15] (see also [24]).

---
**Algorithm 3** Nesterov Algorithm [[16, 15]]

---
1: Init: $w_0 = z_0$. Require: $\theta \leq \frac{1}{L}$.
2: **for** $t = 1, 2, \ldots, T$ **do**
3:     $w_t = z_{t-1} - \theta \nabla f(z_{t-1})$.
4:     $z_t = w_t + \frac{t-1}{t+2}(w_t - w_{t-1})$.
5: **end for**
6: Output $w_T$.

---

To see the equivalence, let us re-write $\bar{x}_t := \frac{1}{A_t}\sum_{s=1}^t \alpha_s x_s$ of Algorithm 2.

$$
\begin{aligned}
\bar{x}_t &= \frac{A_{t-1}\bar{x}_{t-1} + \alpha_t x_t}{A_t} = \frac{A_{t-1}\bar{x}_{t-1} + \alpha_t(x_{t-1} - \gamma_t \alpha_t \nabla f(\tilde{x}_t))}{A_t} \\
&= \frac{A_{t-1}\bar{x}_{t-1} + \alpha_t\left(\frac{A_{t-1}\bar{x}_{t-1} - A_{t-2}\bar{x}_{t-2}}{\alpha_{t-1}} - \gamma_t \alpha_t \nabla f(\tilde{x}_t)\right)}{A_t} \\
&= \bar{x}_{t-1}\left(\frac{A_{t-1}}{A_t} + \frac{\alpha_t(\alpha_{t-1} + A_{t-2})}{A_t \alpha_{t-1}}\right) - \bar{x}_{t-2}\left(\frac{\alpha_t A_{t-2}}{A_t \alpha_{t-1}}\right) - \frac{\gamma_t \alpha_t^2}{A_t}\nabla f(\tilde{x}_t) \qquad (11)\\
&= \bar{x}_{t-1} - \frac{\gamma_t \alpha_t^2}{A_t}\nabla f(\tilde{x}_t) + \left(\frac{\alpha_t A_{t-2}}{A_t \alpha_{t-1}}\right)(\bar{x}_{t-1} - \bar{x}_{t-2}) \\
&= \bar{x}_{t-1} - \frac{1}{4L}\nabla f(\tilde{x}_t) + \left(\frac{t-2}{t+1}\right)(\bar{x}_{t-1} - \bar{x}_{t-2}),
\end{aligned}
$$

where $\alpha_t = t$ and $\gamma_t = \frac{(t+1)}{t}\frac{1}{8L}$.

**Theorem 4.** *Algorithm 3 with $\theta = \frac{1}{4L}$ is equivalent to Algorithm 2 with $\gamma_t = \frac{(t+1)}{t}\frac{1}{8L}$ in the sense that they generate equivalent sequences of iterates:*

$$\text{for all } t = 1, 2, \ldots, T, \qquad w_t = \bar{x}_t \quad \text{and} \quad z_{t-1} = \tilde{x}_t.$$

Let us switch to comparing the update of Algorithm 2, which is (11), with the update of the HEAVY-BALL algorithm. We see that (11) has the so called momentum term (i.e. has a $(\bar{x}_{t-1} - \bar{x}_{t-2})$ term). But, the difference is that the gradient is evaluated at $\tilde{x}_t = \frac{1}{A_t}(\alpha_t x_{t-1} + \sum_{s=1}^{t-1}\alpha_s x_s)$, not $\bar{x}_{t-1} = \frac{1}{A_{t-1}}\sum_{s=1}^{t-1}\alpha_s x_s$, which is the consequence that the y-player plays OPTIMISTICFTL. To

---

**Algorithm 4** HEAVYBALL algorithm

---

1: In the weighted loss setting of Algorithm 1:
2:     $y$-player uses FOLLOWTHELEADER as OAlg$^y$: $y_t = \nabla f(\bar{x}_{t-1})$.
3:     $x$-player uses ONLINEGRADIENTDESCENT as OAlg$^x$:
4:         $x_t := x_{t-1} - \gamma_t \alpha_t \nabla h_t(x) = x_{t-1} - \gamma_t \alpha_t y_t = x_{t-1} - \gamma_t \alpha_t \nabla f(\bar{x}_{t-1})$.

---

elaborate, let us consider a scenario (shown in Algorithm 4) such that the $y$-player plays FOLLOWTHELEADER instead of OPTIMISTICFTL.

Following what we did in (11), we can rewrite $\bar{x}_t$ of Algorithm 4 as

$$\bar{x}_t = \bar{x}_{t-1} - \frac{\gamma_t \alpha_t^2}{A_t} \nabla f(\bar{x}_{t-1}) + (\bar{x}_{t-1} - \bar{x}_{t-2})(\frac{\alpha_t A_{t-2}}{A_t \alpha_{t-1}}), \tag{12}$$

by observing that (11) still holds except that $\nabla f(\widetilde{x}_t)$ is changed to $\nabla f(\bar{x}_{t-1})$ as the y-player uses FOLLOWTHELEADER now, which give us the update of the Heavy Ball algorithm as (12). Moreover, by the regret analysis, we have the following theorem. The proof is in Appendix C.

**Theorem 5.** *Let $\alpha_t = t$. Assume $\mathcal{K} = \mathbb{R}^n$. Also, let $\gamma_t = O(\frac{1}{L})$. The output $\bar{x}_T$ of Algorithm 4 is an $O(\frac{1}{T})$-approximate optimal solution of $\min_x f(x)$.*

To conclude, by comparing Algorithm 2 and Algorithm 4, we see that Nesterov's (1983) method enjoys $O(1/T^2)$ rate since its adopts OPTIMISTICFTL, while the HEAVYBALL algorithm which adopts FTL may not enjoy the fast rate, as the distance terms may not cancel out. The result also conforms to empirical studies that the HEAVYBALL does not exhibit acceleration on general smooth convex problems.

## 4.2 Recovering Nesterov's (1988) 1-memory method [17] and Nesterov's (2005) $\infty$-memory method [19]

In this subsection, we consider recovering Nesterov's (1988) 1-memory method [17] and Nesterov's (2005) $\infty$-memory method [19]. To be specific, we adopt the presentation of Nesterov's algorithm given in Algorithm 1 and Algorithm 3 of [26] respectively.

---

**Algorithm 5** (A) Nesterov's 1-memory method [17] and (B) Nesterov's $\infty$-memory method [19]

---

1: Input: parameter $\beta_t = \frac{2}{t+1}$, $\gamma'_t = \frac{t}{4L}$, $\theta_t = t$, and $\eta = \frac{1}{4L}$.
2: Init: $w_0 = x_0$
3: **for** $t = 1, 2, \ldots, T$ **do**
4:     $z_t = (1 - \beta_t)w_{t-1} + \beta_t x_{t-1}$.
5:     (A) $x_t = \operatorname{argmin}_{x \in \mathcal{K}} \gamma'_t \langle \nabla f(z_t), x \rangle + V_{x_{t-1}}(x)$.
6:     Or, (B) $x_t = \operatorname{argmin}_{x \in \mathcal{K}} \sum_{s=1}^{t} \theta_s \langle x, \nabla f(z_s) \rangle + \frac{1}{\eta} R(x)$, where $R(\cdot)$ is 1-strongly convex.
7:     $w_t = (1 - \beta_t)w_{t-1} + \beta_t x_t$.
8: **end for**
9: Output $w_T$.

---

**Theorem 6.** *Let $\alpha_t = t$. Algorithm 5 with update by option (A) is the case when the y-player uses OPTIMISTICFTL and the x-player adopts MIRRORDESCENT with $\gamma_t = \frac{1}{4L}$ in Fenchel game. Therefore, $w_T$ is an $O(\frac{1}{T^2})$-approximate optimal solution of $\min_{x \in \mathcal{K}} f(x)$.*

The proof is in Appendix D, which shows the direct correspondence of Algorithm 5 using option (A) to our accelerated solution in Section 3.

**Theorem 7.** *Let $\alpha_t = t$. Algorithm 5 with update by option (B) is the case when the y-player uses OPTIMISTICFTL and the x-player adopts BETHEREGULARIZEDLEADER with $\eta = \frac{1}{4L}$ in Fenchel game. Therefore, $w_T$ is an $O(\frac{1}{T^2})$-approximate optimal solution of $\min_{x \in \mathcal{K}} f(x)$.*

The proof is in Appendix E, which requires the regret bound of BETHEREGULARIZEDLEADER.

## 4.3 Accelerated linear rate

Nesterov observed that, when $f(\cdot)$ is both $\mu$-strongly convex and $L$-smooth, one can achieve a rate that is exponentially decaying in $T$ (e.g. page 71-81 of [18]). It is natural to ask if the zero-sum game and regret analysis in the present work also recovers this faster rate in the same fashion. We answer this in the affirmative. Denote $\kappa := \frac{L}{\mu}$. A property of $f(x)$ being $\mu$-strongly convex is that the function $\tilde{f}(x) := f(x) - \frac{\mu\|x\|_2^2}{2}$ is still a convex function. Now we define a new game whose payoff function is $\tilde{g}(x,y) := \langle x, y \rangle - \tilde{f}^*(y) + \frac{\mu\|x\|_2^2}{2}$. Then, the minimax vale of the game is $V^* := \min_x \max_y \tilde{g}(x,y) = \min_x \tilde{f}(x) + \frac{\mu\|x\|_2^2}{2} = \min_x f(x)$. Observe that, in this game, the loss of the y-player in round $t$ is $\alpha_t \ell_t(y) := \alpha_t(\tilde{f}^*(y) - \langle x_t, y \rangle)$, while the loss of the x-player in round $t$ is a strongly convex function $\alpha_t h_t(y) := \alpha_t(\langle x, y_t \rangle + \frac{\mu\|x\|_2^2}{2})$. The cumulative loss function of the x-player becomes more and more strongly convex over time, which is the key to allowing the exponential growth of the total weight $A_t$ that leads to the linear rate. In this setup, we have a "warmup round" $t = 0$, and thus we denote $\tilde{A}_t := \sum_{s=0}^t \alpha_s$ which incorporate the additional step into the average. The proof of the following result is in Appendix H.

**Theorem 8.** *For the game $\tilde{g}(x,y) := \langle x, y \rangle - \tilde{f}^*(y) + \frac{\mu\|x\|_2^2}{2}$, if the y-player plays* OPTIMISTICFTL *and the x-player plays* BETHEREGULARIZEDLEADER: $x_t \leftarrow \arg\min_{x \in \mathcal{X}} \sum_{s=0}^t \alpha_s \ell_s(x)$, where $\alpha_0 \ell_0(x) := \alpha_0 \frac{\mu\|x\|_2^2}{2}$, then the weighted average points $(\bar{x}_T, \bar{y}_T)$ would be an $O(\exp(-\frac{T}{\sqrt{\kappa}}))$- approximate equilibrium of the game, where the weights $\alpha_0, \alpha_1, \ldots$ are chosen to satisfy $\frac{\alpha_t}{\tilde{A}_t} = \frac{1}{\sqrt{6\kappa}}$. This implies that $f(\bar{x}_T) - \min_{x \in \mathcal{X}} f(x) = O(\exp(-\frac{T}{\sqrt{\kappa}}))$.*

## 5 Accelerated Proximal Method

In this section, we consider solving composite optimization problems $\min_{x \in \mathbb{R}^n} f(x) + \psi(x)$, where $f(\cdot)$ is smooth convex but $\psi(\cdot)$ is possibly non-differentiable convex (e.g. $\|\cdot\|_1$). We want to show that the game analysis still applies to this problem. We just need to change the payoff function $g$ to account for $\psi(x)$. Specifically, we consider the following two-players zero-sum game, $\min_x \max_y \{\langle x, y \rangle - f^*(y) + \psi(x)\}$. Notice that the minimax value of the game is $\min_x f(x) + \psi(x)$, which is exactly the optimum value of the composite optimization problem. Let us denote the proximal operator as $\textbf{prox}_{\lambda\psi}(v) = \arg\min_x \left(\psi(x) + \frac{1}{2\lambda}\|x - v\|_2^2\right)$. [1]

---

**Algorithm 6** Accelerated Proximal Method

---
1: In the weighted loss setting of Algorithm 1 (let $\alpha_t = t$ and $\gamma_t = \frac{1}{4L}$):
2:     y-player uses OPTIMISITCFTL as OAlg$^y$: $y_t = \nabla f(\tilde{x}_t)$.
3:     x-player uses MIRRORDESCENT with $\psi(x) := \frac{1}{2}\|x\|_2^2$ in Bregman divergence as OAlg$^x$:
4:       $x_t = \arg\min_x \gamma_t(\alpha_t h_t(x)) + V_{x_{t-1}}(x) = \arg\min_x \gamma_t(\alpha_t\{\langle x, y_t \rangle + \psi(x)\}) + V_{x_{t-1}}(x)$
5:       $= \arg\min_x \phi(x) + \frac{1}{2\alpha_t\gamma_t}(\|x\|_2^2 + 2\langle\alpha_t\gamma_t y_t - x_{t-1}, x\rangle) = \textbf{prox}_{\alpha_t\gamma_t\psi}(x_{t-1} - \alpha_t\gamma_t\nabla f(\tilde{x}_t))$

---

We notice that the loss function of the x-player here, $\alpha_t h_t(x) = \alpha_t(\langle x, y_t \rangle + \psi(x))$, is possibly nonlinear. Yet, we can slightly adapt the analysis in Section 3 to show that the weighed average $\bar{x}_T$ is still an $O(1/T^2)$ approximate optimal solution of the offline problem. Please see Appendix I for details. One can view Algorithm 6 as a variant of the so called "Accelerated Proximal Gradient" in [6]. Yet, the design and analysis of our algorithm is simpler than that of [6].

**Acknowlegement:** We would like to thank Kevin Lai and Kfir Levy for helpful discussions leading up to the results in this paper. This work was supported by funding from the Division of Computer Science and Engineering at the University of Michigan, from the College of Computing at the Georgia Institute of Technology, NSF TRIPODS award 1740776, and NSF CAREER award 1453304.

## Footnotes

[1]It is known that for some $\psi(\cdot)$, their corresponding proximal operations have closed-form solutions (see e.g. [20] for details).

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
