[Supplementary Material]

# Acceleration through Optimistic No-Regret Dynamics (Appendix)

**Jun-Kun Wang**
College of Computing
Georgia Institute of Technology
Atlanta, GA 30313
jimwang@gatech.edu

**Jacob Abernethy**
College of Computing
Georgia Institute of Technology
Atlanta, GA 30313
prof@gatech.edu

## A  Two key lemmas

**Lemma 4** *Let the sequence of $x_t$'s be chosen according to* MIRRORDESCENT. *Assume that the Bregman Divergence is uniformly bounded on $\mathcal{K}$, so that $D = \sup_{t=1,\ldots,T} V_{x_t}(x^*)$, where $x^*$ denotes the minimizer of $f(\cdot)$. Assume that the sequence $\{\gamma_t\}_{t=1,2,\ldots}$ is non-increasing. Then we have* $\boldsymbol{\alpha}\text{-}\text{REG}^x \leq \frac{D}{\gamma_T} - \sum_{t=1}^{T} \frac{1}{2\gamma_t}\|x_{t-1} - x_t\|^2$.

*Proof.*  The key inequality we need, which can be found in Lemma 1 of [5] (and for completeness is included in Appendix A) is as follows: let $y, c$ be arbitrary, and assume $x^+ = \operatorname{argmin}_{x \in \mathcal{K}}\langle x, y\rangle + V_c(x)$, then for any $x^* \in \mathcal{K}$, $\langle x^+ - x^*, y\rangle \leq V_c(x^*) - V_{x^+}(x^*) - V_c(x^+)$. Now apply this fact for $x^+ = x_t, y = \gamma_t \alpha_t y_t$ and $c = x_{t-1}$, which provides

$$\langle x_t - x^*, \gamma_t \alpha_t y_t\rangle \leq V_{x_{t-1}}(x^*) - V_{x_t}(x^*) - V_{x_{t-1}}(x_t). \tag{1}$$

So, the weighted regret of the x-player can be bounded by

$$
\begin{aligned}
\boldsymbol{\alpha}\text{-}\text{REG}^x &:= \sum_{t=1}^{T} \alpha_t \langle x_t - x^*, y_t\rangle \overset{(1)}{\leq} \sum_{t=1}^{T} \frac{1}{\gamma_t}\big(V_{x_{t-1}}(x^*) - V_{x_t}(x^*) - V_{x_{t-1}}(x_t)\big) \\
&= \frac{1}{\gamma_1} V_{x_0}(x^*) - \frac{1}{\gamma_T} v_{x_T}(x^*) + \sum_{t=1}^{T-1}\big(\frac{1}{\gamma_{t+1}} - \frac{1}{\gamma_t}\big)V_{x_t}(x^*) - \frac{1}{\gamma_t}V_{x_{t-1}}(x_t) \\
&\overset{(a)}{\leq} \frac{1}{\gamma_1}D + \sum_{t=1}^{T-1}\big(\frac{1}{\gamma_{t+1}} - \frac{1}{\gamma_t}\big)D - \frac{1}{\gamma_t}V_{x_{t-1}}(x_t) = \frac{D}{\gamma_T} - \sum_{t=1}^{T} \frac{1}{\gamma_t}V_{x_{t-1}}(x_t) \\
&\overset{(b)}{\leq} \frac{D}{\gamma_T} - \sum_{t=1}^{T} \frac{1}{2\gamma_t}\|x_{t-1} - x_t\|^2,
\end{aligned}
\tag{2}
$$

where $(a)$ holds since the sequence $\{\gamma_t\}$ is non-increasing and $D$ upper bounds the divergence terms, and $(b)$ follows from the strong convexity of $\phi$, which grants $V_{x_{t-1}}(x_t) \geq \frac{1}{2}\|x_t - x_{t-1}\|^2$.  $\square$

The above lemma requires a bound $D$ on the divergence terms $V_{x_t}(x^*)$, which might be large in certain unconstrained settings – recall that we do no necessarily require that $\mathcal{K}$ is a bounded set, we only assume that $f(\cdot)$ is minimized at a point with finite norm. On the other hand, when the x-player's learning rate $\gamma$ is fixed, we can define the more natural choice $D = V_{x_0}(x^*)$.

**Lemma 4 [Alternative]:**  *Let the sequence of $x_t$'s be chosen according to* MIRRORDESCENT, *and assume $\gamma_t = \gamma$ for all $t$. Let $D = V_{x_0}(x^*)$, where $x^*$ denotes the benchmark in $\boldsymbol{\alpha}\text{-}\text{REG}^x$. Then we have $\boldsymbol{\alpha}\text{-}\text{REG}^x \leq \frac{D}{\gamma} - \sum_{t=1}^{T} \frac{1}{2\gamma}\|x_{t-1} - x_t\|^2$.*

*Proof.*  The proof follows exactly as before, yet $\gamma_t = \gamma_{t+1}$ for all $t$ implies that $\frac{1}{\gamma_{t+1}} - \frac{1}{\gamma_t} = 0$ and we may drop the sum in the third line of (2). The rest of the proof is identical.  $\square$

**Lemma 1 of [5]**: *Let $x' = \arg\min_{x\in\mathcal{K}} \langle x, y\rangle + V_c(x)$. Then, it satisfies that for any $x^* \in \mathcal{K}$,*

$$\langle x' - x^*, y\rangle \leq V_c(x^*) - V_{x'}(x^*) - V_c(x'). \tag{3}$$

*Proof.* Recall that the Bregman divergence with respect to the distance generating function $\phi(\cdot)$ at a point $c$ is: $V_c(x) := \phi(x) - \langle\nabla\phi(c), x - c\rangle - \phi(c)$.

Denote $F(x) := \langle x, y\rangle + V_c(x)$. Since $x'$ is the optimal point of $\arg\min_{x\in K} F(x)$, by optimality, $\langle x^* - x', \nabla F(x')\rangle \geq 0$, for any $x^* \in K$. So,

$$
\begin{aligned}
\langle x^* - x', \nabla F(x')\rangle &= \langle x^* - x', y\rangle + \langle x^* - x', \nabla\phi(x') - \nabla\phi(c)\rangle \\
&= \langle x^* - x', y\rangle + \{\phi(x^*) - \langle\nabla\phi(c), x^* - c\rangle - \phi(c)\} - \{\phi(x^*) - \langle\nabla\phi(x'), x^* - x'\rangle - \phi(x')\} \\
&\quad - \{\phi(x') - \langle\nabla\phi(c), x' - c\rangle - \phi(c)\} \\
&= \langle x^* - x', y\rangle + V_c(x^*) - V_{x'}(x^*) - V_c(x') \geq 0.
\end{aligned}
\tag{4}
$$

The last inequality means that

$$\langle x' - x^*, y\rangle \leq V_c(x^*) - V_{x'}(x^*) - V_c(x'). \tag{5}$$

$\square$

## B  Proof of Theorem 4

**Theorem 4** *Algorithm 3 with $\theta = \frac{1}{4L}$ is equivalent to Algorithm 2 with $\gamma_t = \frac{(t+1)}{t}\frac{1}{8L}$ in the sense that they generate equivalent sequences of iterates:*

$$\text{for all } t = 1, 2, \ldots, T, \qquad w_t = \bar{x}_t \quad \text{and} \quad z_{t-1} = \widetilde{x}_t.$$

*Proof.* First, let us check the base case to see if $w_1 = \bar{x}_1$. We have that $w_1 = z_0 - \theta\nabla f(z_0)$ from line 3 of Algorithm 3, while $\bar{x}_1 = \bar{x}_0 - \frac{1}{4L}\nabla f(\widetilde{x}_1)$ in (11). Thus, if the initialization is the same: $w_0 = z_0 = x_0 = \bar{x}_0 = \widetilde{x}_1$, then $w_1 = \bar{x}_1$.

Now assume that $w_{t-1} = \bar{x}_{t-1}$ holds for a $t \geq 2$. Then, from the expression of line 4 that $z_{t-1} = w_{t-1} + \frac{t-2}{t+1}(w_{t-1} - w_{t-2})$, we get $z_{t-1} = \bar{x}_{t-1} + \frac{t-2}{t+1}(\bar{x}_{t-1} - \bar{x}_{t-2})$. Let us analyze that the r.h.s of the equality. The coefficient of $x_{t-1}$ in $\bar{x}_{t-1} + \frac{t-2}{t+1}(\bar{x}_{t-1} - \bar{x}_{t-2})$ is $\frac{(t-1)+\frac{t-2}{t+1}(t-1)}{A_{t-1}} = \frac{2(1+\frac{t-2}{t+1})}{t} = \frac{2(2t-1)}{t(t+1)}$, while the coefficient of each $x_\tau$ for any $\tau \leq t-2$ in $\bar{x}_{t-1} + \frac{t-2}{t+1}(\bar{x}_{t-1} - \bar{x}_{t-2})$ is $\frac{(1+\frac{t-2}{t+1})\tau}{A_{t-1}} - \frac{t-2}{t+1}\frac{\tau}{A_{t-2}} = \{\frac{2(2t-1)}{(t-1)t(t+1)} - \frac{2}{(t+1)(t-1)}\} \times \tau = \{\frac{2}{(t-1)(t+1)}(\frac{2t-1}{t} - 1)\} \times \tau = \frac{2\tau}{t(t+1)}$. Yet, the coefficient of $x_{t-1}$ in $\widetilde{x}_t$ is $\frac{t+(t-1)}{A_t} = \frac{2(2t-1)}{t(t+1)}$ and the coefficient of $x_\tau$ in $\widetilde{x}_t$ is $\frac{\tau}{A_t} = \frac{2\tau}{t(t+1)}$ for any $\tau \leq t-2$. Thus, $z_{t-1} = \widetilde{x}_t$. Now observe that if $z_{t-1} = \widetilde{x}_t$, we get $w_t = \bar{x}_t$. To see this, substituting $z_{t-1} = w_{t-1} + \frac{t-2}{t+1}(w_{t-1} - w_{t-2})$ of line 4 into line 3, we get $w_t = w_{t-1} + \frac{t-2}{t+1}(w_{t-1} - w_{t-2}) - \theta\nabla f(z_{t-1})$. By using $z_{t-1} = \widetilde{x}_t$ and $w_{t-1} = \bar{x}_{t-1}$, we further get $w_t = \bar{x}_{t-1} + \frac{t-2}{t+1}(\bar{x}_{t-1} - \bar{x}_{t-2}) - \theta\nabla f(\widetilde{x}_t) = \bar{x}_t$. We can repeat the argument to show that the correspondence holds for any $t$, which establishes the equivalency.

Notice that the choice of decreasing sequence $\{\gamma_t\}$ here can still make the distance terms in (10) cancel out. So, we get $O(1/T^2)$ rate by the guarantee. $\square$

## C  Proof of Theorem 5

**Theorem 5** *Let $\alpha_t = t$. Assume $\mathcal{K} = \mathbb{R}^n$. Also, let $\gamma_t = O(\frac{1}{L})$. The output $\bar{x}_T$ of Algorithm 4 is an $O(\frac{1}{T})$-approximate optimal solution of $\min_x f(x)$.*

*Proof.* To analyze the guarantee of $\bar{x}_T$ of Algorithm 4, we use the following lemma about FOLLOWTHELEADER for strongly convex loss functions.

**Corollary 1 from [3]** *Let $\ell_1, ..., \ell_T$ be a sequence of functions such that for all $t \in [T]$, $\ell_t$ is $\sigma_t$-strongly convex. Assume that* FOLLOWTHELEADER *runs on this sequence and for each $t \in [T]$, let $\theta_t$ be in $\nabla \ell_t(y_t)$. Then, $\sum_{t=1}^T \ell_t(y_t) - \min_x \sum_{t=1}^T \ell_t(y) \leq \frac{1}{2} \sum_{t=1}^T \frac{\|\theta_t\|^2}{\sum_{\tau=1}^t \sigma_\tau}$*

Observe that the $y$-player plays FOLLOWTHELEADER on the loss function sequence $\alpha_t \ell_t(y) := \alpha_t(-\langle x_t, y \rangle + f^*(y))$, whose strong convexity parameter is $\frac{\alpha_t}{L}$ (due to $f^*(y)$ is $\frac{1}{L}$-strongly convex by duality). Also, $\nabla \ell_t(y_t) = -x_t + \nabla f^*(y_t) = -x_t + \bar{x}_{t-1}$, where the last inequality is due to that if $y_t = \mathrm{argmax}_y \langle \frac{1}{A_{t-1}} \sum_{s=1}^{t-1} \alpha_s x_s, y \rangle - f^*(y) = \nabla f(\bar{x}_{t-1})$, then $\bar{x}_{t-1} = \nabla f^*(y_t)$ by duality. So, we have $\overline{\boldsymbol{\alpha}\text{-REG}}^y \overset{AboveCor.}{\leq} \frac{1}{2A_T} \sum_{t=1}^T \frac{\alpha_t^2 \|\bar{x}_{t-1} - x_t\|^2}{\sum_{\tau=1}^t \alpha_\tau (1/L)} = \frac{1}{2A_T} \sum_{t=1}^T \frac{\alpha_t^2 L \|\bar{x}_{t-1} - x_t\|^2}{A_t} = O(\sum_{\tau=1}^T \frac{L \|\bar{x}_{t-1} - x_t\|^2}{A_T})$. For the $x$-player, it is an instance of MIRRORDESCENT, so $\overline{\boldsymbol{\alpha}\text{-REG}}^x := \frac{1}{A_T} \sum_{t=1}^T \langle x_t - x^*, \alpha_t y_t \rangle \leq \frac{\frac{1}{\gamma_T} D - \sum_{t=1}^T \frac{1}{2\gamma_t} \|x_{t-1} - x_t\|^2}{A_T}$ Therefore, $\bar{x}_T$ of Algorithm 4 is an $\overline{\boldsymbol{\alpha}\text{-REG}}^x + \overline{\boldsymbol{\alpha}\text{-REG}}^y = O(\frac{L \sum_{t=1}^T (\|\bar{x}_{t-1} - x_t\|^2 - \|x_t - x_{t-1}\|^2)}{A_T})$ -approximate optimal solution. Since the distance terms may not cancel out, one may only bound the differences of the distance terms by a constant, which leads to the non-accelerated $O(1/T)$ rate. $\quad\square$

## D  Proof of Theorem 6

**Theorem 6** *Let $\alpha_t = t$. Algorithm 5 with update by option (A) is the case when the y-player uses* OPTIMISTICFTL *and the x-player adopts* MIRRORDESCENT *with $\gamma_t = \frac{1}{4L}$ in Fenchel game. Therefore, $w_T$ is an $O(\frac{1}{T^2})$-approximate optimal solution of $\min_{x \in \mathcal{K}} f(x)$.*

*Proof.* We first prove by induction showing that $w_t$ in Algorithm 5 is $\sum_{s=1}^t \frac{\alpha_s}{A_t} x_s$ for any $t > 0$. For the base case $t = 1$, we have $w_1 = (1 - \beta_1) w_0 + \beta_1 x_1 = x_1 = \frac{\alpha_1}{A_1} x_1$. Now suppose that the equivalency holds at $t - 1$, for a $t \geq 2$. Then,

$$
\begin{aligned}
w_t &= (1 - \beta_t) w_{t-1} + \beta_t x_t \overset{(a)}{=} (1 - \beta_t)(\sum_{s=1}^{t-1} \frac{\alpha_s}{A_{t-1}} x_s) + \beta_t x_t \\
&= (1 - \frac{2}{t+1})(\sum_{s=1}^{t-1} \frac{\alpha_s}{\frac{t(t-1)}{2}} x_s) + \beta_t x_t = \sum_{s=1}^{t-1} \frac{\alpha_s}{\frac{t(t+1)}{2}} x_s + \frac{\alpha_t}{A_t} x_t = \sum_{s=1}^t \frac{\alpha_s}{A_s} x_s,
\end{aligned}
\tag{6}
$$

where $(a)$ is by induction. So, it holds at $t$ too. Now we are going to show that $z_t = \frac{1}{A_t}(\alpha_t x_{t-1} + \sum_{s=1}^{t-1} \alpha_s x_s) = \tilde{x}_t$. We have that $z_t = (1 - \beta_t) w_{t-1} + \beta_t x_{t-1} = (1 - \beta_t)(\sum_{s=1}^{t-1} \frac{\alpha_s}{A_{t-1}} x_s) + \beta_t x_{t-1} = (1 - \frac{2}{t+1})(\sum_{t=1}^{t-1} \frac{\alpha_s}{\frac{t(t-1)}{2}} x_t) + \beta_t x_{t-1} = \sum_{s=1}^{t-1} \frac{\alpha_s}{\frac{t(t+1)}{2}} x_s + \beta_t x_{t-1} = \sum_{s=1}^{t-1} \frac{\alpha_s}{A_t} x_s + \frac{\alpha_t}{A_t} x_{t-1} = \tilde{x}_t$. The result also means that $\nabla f(z_t) = \nabla f(\tilde{x}_t) = y_t$ of the y-player who plays `Optimistic-FTL` in Algorithm 1. Furthermore, it shows that line 5 of Algorithm 5: $x_t = \mathrm{argmin}_{x \in \mathcal{K}} \gamma_t' \langle \nabla f(z_t), x \rangle + V_{x_{t-1}}(x)$ is exactly (9) of MIRRORDESCENT in *Fenchel game*. Also, from (6), the last iterate $w_T$ in Algorithm 5 corresponds to the final output of our accelerated solution to *Fenchel game*, which is the weighted average point that enjoys the guarantee by the game analysis. $\quad\square$

## E  Proof of Theorem 7

**Theorem 7** *Let $\alpha_t = t$. Algorithm 5 with update by option (B) is the case when the y-player uses* OPTIMISTICFTL *and the x-player adopts* BETHEREGULARIZEDLEADER *with $\eta = \frac{1}{4L}$ in Fenchel game. Therefore, $w_T$ is an $O(\frac{1}{T^2})$-approximate optimal solution of $\min_{x \in \mathcal{K}} f(x)$.*

*Proof.* Consider in *Fenchel game* that the y-player uses OPTIMISTICFTL while the x-player plays according to BTRL:

$$
x_t = \mathrm{argmin}_{x \in \mathcal{K}} \sum_{t=1}^T \langle x_t, \alpha_t y_t \rangle + \frac{1}{\eta} R(x),
$$

where $R(\cdot)$ is a 1-strongly convex function. Define, $z = \arg\min_{x \in \mathcal{K}} R(x)$. Form [1] (also see Appendix F), it shows that BTRL has regret

$$\text{Regret} := \sum_{t=1}^{T} \langle x_t - x^*, \alpha_t y_t \rangle \leq \frac{R(x^*) - R(z) - \frac{1}{2}\sum_{t=1}^{T}\|x_t - x_{t-1}\|^2}{\eta}, \tag{7}$$

where $x^*$ is the benchmark/comparator defined in the definition of the weighted regret (4).

By combining (8) and (7), we get that

$$\frac{\boldsymbol{\alpha}\text{-}\text{REG}^x + \boldsymbol{\alpha}\text{-}\text{REG}^y}{A_T} = \frac{\frac{R(x^*)-R(z)}{\eta} + \sum_{t=1}^{T}(\frac{\alpha_t^2}{A_t}L - \frac{1}{2\eta})\|x_{t-1}-x_t\|^2}{A_T} \leq O\left(\frac{L(R(x^*)-R(z))}{T^2}\right), \tag{8}$$

where the last inequality is because $\eta = \frac{1}{4L}$ so that the distance terms cancel out. So, by Lemma 1 and Theorem 1 again, we know that $\bar{x}_T$ is an $O(\frac{1}{T^2})$-approximate optimal solution of $\min_{x \in \mathcal{K}} f(x)$.

The remaining thing to do is showing that $\bar{x}_T$ is actually $w_T$ of Algorithm 5 with option (B). But, this follows the same line as the proof of Theorem 6. So, we have completed the proof. $\square$

# F  Proof of BETHEREGULARIZEDLEADER 's regret

For completeness, we replicate the proof in [1] about the regret bound of BETHEREGULAR-IZEDLEADER in this section.

**Theorem 10** of [[1]]  *Let $\theta_t$ be the loss vector in round $t$. Let the update of BTRL be $x_t = \arg\min_{x \in \mathcal{K}} \langle x, L_t \rangle + \frac{1}{\eta} R(x)$, where $R(\cdot)$ is $\beta$-strongly convex. Denote $z = \arg\min_{x \in \mathcal{K}} R(x)$. Then, BTRL has regret*

$$\text{Regret} := \sum_{t=1}^{T} \langle x_t - x^*, \theta_t \rangle \leq \frac{R(x^*) - R(z) - \frac{\beta}{2}\sum_{t=1}^{T}\|x_t - x_{t-1}\|^2}{\eta}. \tag{9}$$

To analyze the regret of BETHEREGULARIZEDLEADER, let us consider OPTIMISTICFTRL first. Let $\theta_t$ be the loss vector in round $t$ and let the cumulative loss vector be $L_t = \sum_{s=1}^{t} \theta_s$. The update of OPTIMISTICFTRL is

$$x_t = \arg\min_{x \in \mathcal{K}} \langle x, L_{t-1} + m_t \rangle + \frac{1}{\eta} R(x), \tag{10}$$

where $m_t$ is the learner's guess of the loss vector in round $t$, $R(\cdot)$ is $\beta$-strong convex with respect to a norm $(\|\cdot\|)$ and $\eta$ is a parameter. Therefore, it is clear that the regret of BETHEREGULARIZEDLEADER will be the one when OPTIMISTICFTRL 's guess of the loss vectors exactly match the true ones, i.e. $m_t = \theta_t$.

**Theorem 16** of [[1]]  *Let $\theta_t$ be the loss vector in round $t$. Let the update of OPTIMISTICFTRL be $x_t = \arg\min_{x \in \mathcal{K}} \langle x, L_{t-1} + m_t \rangle + \frac{1}{\eta} R(x)$, where $m_t$ is the learner's guess of the loss vector in round $t$ and $R(x)$ is a $\beta$-strongly convex function. Denote the update of standard FTRL as $z_t = \arg\min_{x \in \mathcal{K}} \langle x, L_{t-1} \rangle + \frac{1}{\eta} R(x)$. Also, $z_1 = \arg\min_{x \in \mathcal{K}} R(x)$. Then, OPTIMISTICFTRL (10) has regret*

$$\text{Regret} := \sum_{t=1}^{T} \langle x_t - x^*, \theta_t \rangle \leq \frac{R(x^*) - R(z_1) - D_T}{\eta} + \sum_{t=1}^{T} \frac{\eta}{\beta}\|\theta_t - m_t\|_*^2, \tag{11}$$

*where $D_T = \sum_{t=1}^{T} \frac{\beta}{2}\|x_t - z_t\|^2 + \frac{\beta}{2}\|x_t - z_{t+1}\|^2$, $z_t = \arg\min_{x \in \mathcal{K}} \langle x, L_{t-1} \rangle + \frac{1}{\eta} R(x)$, and $x_t = \arg\min_{x \in \mathcal{K}} \langle x, L_{t-1} + m_t \rangle + \frac{1}{\eta} R(x)$.*

Recall that the update of BETHEREGULARIZEDLEADER is $x_t = \arg\min_{x \in \mathcal{K}} \langle x, L_t \rangle + \frac{1}{\eta} R(x)$, Therefore, we have that $m_t = \theta_t$ and $x_t = z_{t+1}$ in the regret bound of OPTIMISTICFTRL indicated by the theorem. Consequently, we get that the regret of BETHEREGULARIZEDLEADER satisfies

$$\text{Regret} := \sum_{t=1}^{T} \langle x_t - x^*, \theta_t \rangle \leq \frac{R(x^*) - R(z) - \frac{\beta}{2}\sum_{t=1}^{T}\|x_t - x_{t-1}\|^2}{\eta}. \tag{12}$$

# G  Proof of OPTIMISTICFTRL 's regret

For completeness, we replicate the proof in [1] about the regret bound of OPTIMISTICFTRL in this section.

**Theorem 16** of [[1]] *Let $\theta_t$ be the loss vector in round t. Let the update of OPTIMISTICFTRL be $x_t = \arg\min_{x \in \mathcal{K}} \langle x, L_{t-1} + m_t \rangle + \frac{1}{\eta} R(x)$, where $m_t$ is the learner's guess of the loss vector in round t and $R(x)$ is a $\beta$-strongly convex function. Denote the update of standard FTRL as $z_t = \arg\min_{x \in \mathcal{K}} \langle x, L_{t-1} \rangle + \frac{1}{\eta} R(x)$. Also, $z_1 = \arg\min_{x \in \mathcal{K}} R(x)$. Then, OPTIMISTICFTRL (10) has regret*

$$\text{Regret} := \sum_{t=1}^{T} \langle x_t - x^*, \theta_t \rangle \leq \frac{R(x^*) - R(z_1) - D_T}{\eta} + \sum_{t=1}^{T} \frac{\eta}{\beta} \|\theta_t - m_t\|_*^2, \qquad (13)$$

*where $D_T = \sum_{t=1}^{T} \frac{\beta}{2} \|x_t - z_t\|^2 + \frac{\beta}{2} \|x_t - z_{t+1}\|^2$, $z_t = \arg\min_{x \in \mathcal{K}} \langle x, L_{t-1} \rangle + \frac{1}{\eta} R(x)$, and $x_t = \arg\min_{x \in \mathcal{K}} \langle x, L_{t-1} + m_t \rangle + \frac{1}{\eta} R(x)$.*

*Proof.* Define $z_t = \arg\min_{x \in \mathcal{K}} \langle x, L_{t-1} \rangle + \frac{1}{\eta} R(x)$ as the update of the standard FOLLOW-THE-REGULARIZED-LEADER. We can re-write the regret as

$$\text{Regret} := \sum_{t=1}^{T} \langle x_t - x^*, \theta_t \rangle = \sum_{t=1}^{T} \langle x_t - z_{t+1}, \theta_t - m_t \rangle + \sum_{t=1}^{T} \langle x_t - z_{t+1}, m_t \rangle + \langle z_{t+1} - x^*, \theta_t \rangle \tag{14}$$

Let us analyze the first sum

$$\sum_{t=1}^{T} \langle x_t - z_{t+1}, \theta_t - m_t \rangle. \tag{15}$$

Now using Lemma 17 of [1] (which is also stated below) with $x_1 = x_t$, $u_1 = \sum_{s=1}^{t-1} \theta_s + m_t$ and $x_2 = z_{t+1}$, $u_2 = \sum_{s=1}^{t} \theta_s$ in the lemma, we have

$$\sum_{t=1}^{T} \langle x_t - z_{t+1}, \theta_t - m_t \rangle \leq \sum_{t=1}^{T} \|x_t - z_{t+1}\| \|\theta_t - m_t\|_* \leq \sum_{t=1}^{T} \frac{\eta}{\beta} \|\theta_t - m_t\|_*^2. \tag{16}$$

For the other sum,

$$\sum_{t=1}^{T} \langle x_t - z_{t+1}, m_t \rangle + \langle z_{t+1} - x^*, \theta_t \rangle, \tag{17}$$

we are going to show that, for any $T \geq 0$, it is upper-bounded by $\frac{R(x^*) - R(z_1) - D_T}{\eta}$, which holds for any $x^* \in \mathcal{K}$, where $D_T = \sum_{t=1}^{T} \frac{\beta}{2} \|x_t - z_t\|^2 + \frac{\beta}{2} \|x_t - z_{t+1}\|^2$. For the base case $T = 0$, we see that

$$\sum_{t=1}^{0} \langle x_t - z_{t+1}, m_t \rangle + \langle z_{t+1} - x^*, \theta_t \rangle = 0 \leq \frac{R(x^*) - R(z_1) - 0}{\eta}, \tag{18}$$

as $z_1 = \arg\min_{x \in \mathcal{K}} R(x)$.

Using induction, assume that it also holds for $T - 1$ for a $T \geq 1$. Then, we have

$$
\begin{aligned}
&\sum_{t=1}^{T} \langle x_t - z_{t+1}, m_t \rangle + \langle z_{t+1}, \theta_t \rangle \\
&\overset{(a)}{\leq} \langle x_T - z_{T+1}, m_T \rangle + \langle z_{T+1}, \theta_T \rangle + \frac{R(z_T) - R(z_1) - D_{T-1}}{\eta} + \langle z_T, L_{T-1} \rangle \\
&\overset{(b)}{\leq} \langle x_T - z_{T+1}, m_T \rangle + \langle z_{T+1}, \theta_T \rangle + \frac{R(x_T) - R(z_1) - D_{T-1} - \frac{\beta}{2} \|x_T - z_T\|^2}{\eta} + \langle x_T, L_{T-1} \rangle \\
&= \langle z_{T+1}, \theta_T - m_T \rangle + \frac{R(x_T) - R(z_1) - D_{T-1} - \frac{\beta}{2} \|x_T - z_T\|^2}{\eta} + \langle x_T, L_{T-1} + m_T \rangle \\
&\overset{(c)}{\leq} \langle z_{T+1}, \theta_T - m_T \rangle + \frac{R(z_{T+1}) - R(z_1) - D_{T-1} - \frac{\beta}{2} \|x_T - z_T\|^2 - \frac{\beta}{2} \|x_T - z_{T+1}\|^2}{\eta} \\
&\quad + \langle z_{T+1}, L_{T-1} + m_T \rangle \\
&= \langle z_{T+1}, L_T \rangle + \frac{R(z_{T+1}) - R(z_1) - D_T}{\eta} \\
&\overset{(d)}{\leq} \langle x^*, L_T \rangle + \frac{R(x^*) - R(z_1) - D_T}{\eta},
\end{aligned}
\tag{19}
$$

where (a) is by induction such that the inequality holds at $T - 1$ for any $x^* \in \mathcal{K}$ including $x^* = z_T$, (b) and (c) are by strong convexity so that

$$\langle z_T, L_{T-1} \rangle + \frac{R(z_T)}{\eta} \leq \langle x_T, L_{T-1} \rangle + \frac{R(x_T)}{\eta} - \frac{\beta}{2\eta} \|x_T - z_T\|^2, \tag{20}$$

and

$$\langle x_T, L_{T-1} + m_T \rangle + \frac{R(x_T)}{\eta} \leq \langle z_{T+1}, L_{T-1} + m_T \rangle + \frac{R(z_{T+1})}{\eta} - \frac{\beta}{2\eta} \|x_T - z_{T+1}\|^2, \tag{21}$$

and (d) is because $z_{T+1}$ is the optimal point of $\arg\min_x \langle x, L_T \rangle + \frac{R(x)}{\eta}$. We've completed the induction.

$\square$

**Lemma 17 of [[1]]** *Denote $x_1 = \arg\min_x \langle x, u_1 \rangle + \frac{1}{\eta} R(x)$ and $x_2 = \arg\min_x \langle x, u_2 \rangle + \frac{1}{\eta} R(x)$ for a $\beta$-strongly convex function $R(\cdot)$ with respect to a norm $\|\cdot\|$. We have $\|x_1 - x_2\| \leq \frac{\eta}{\beta} \|u_1 - u_2\|_*$.*

## H    Proof of Theorem 8

**Theorem 8** *For the game $g(x,y) := \langle x, y \rangle - \tilde{f}^*(y) + \frac{\mu \|x\|_2^2}{2}$, if the y-player plays OPTIMISTICFTL and the x-player plays BETHEREGULARIZEDLEADER: $x_t \leftarrow \arg\min_{x \in \mathcal{X}} \sum_{s=0}^{t} \alpha_s \ell_s(x)$, where $\alpha_0 \ell_0(x) := \alpha_0 \frac{\mu \|x\|_2^2}{2}$, then the weighted average $(\bar{x}_T, \bar{y}_T)$ would be $O(\exp(-\frac{T}{\sqrt{\kappa}}))$-approximate equilibrium of the game, where the weights $\frac{\alpha_t}{\tilde{A}_t} = \frac{1}{\sqrt{6\kappa}}$. This implies that $f(\bar{x}_T) - \min_{x \in \mathcal{X}} f(x) = O(\exp(-\frac{T}{\sqrt{\kappa}}))$.*

*Proof.* From Lemma 3, we know that the y-player's regret by OPTIMISTICFTL is

$$
\begin{aligned}
\sum_{t=1}^{T} \alpha_t \ell_t(\tilde{y}_t) - \alpha_t \ell_t(y^*) &\leq \sum_{t=1}^{T} \delta_t(\tilde{y}_t) - \delta_t(\hat{y}_{t+1}) \\
&= \sum_{t=1}^{T} \alpha_t \langle x_{t-1} - x_t, \tilde{y}_t - \hat{y}_{t+1} \rangle \\
\text{(Eqns. 5, 6)} \quad &= \sum_{t=1}^{T} \alpha_t \langle x_{t-1} - x_t, \nabla \tilde{f}(\tilde{x}_t) - \nabla \tilde{f}(\bar{x}_t) \rangle \\
\text{(Hölder's Ineq.)} \quad &\leq \sum_{t=1}^{T} \alpha_t \|x_{t-1} - x_t\| \|\nabla \tilde{f}(\tilde{x}_t) - \nabla \tilde{f}(\bar{x}_t)\| \\
&= \sum_{t=1}^{T} \alpha_t \|x_{t-1} - x_t\| \|\nabla f(\tilde{x}_t) - \mu \tilde{x}_t - \nabla \tilde{f}(\bar{x}_t) + \mu \bar{x}_t\| \\
\text{(triangle inequality)} \quad &\leq \sum_{t=1}^{T} \alpha_t \|x_{t-1} - x_t\| (\|\nabla f(\tilde{x}_t) - \nabla \tilde{f}(\bar{x}_t)\| + \mu \|\bar{x}_t - \tilde{x}_t\|) \\
\text{($L$-smoothness and $L \geq \mu$)} \quad &\leq 2L \sum_{t=1}^{T} \alpha_t \|x_{t-1} - x_t\| \|\tilde{x}_t - \bar{x}_t\| \\
\text{(Eqn. 7)} \quad &= 2L \sum_{t=1}^{T} \frac{\alpha_t^2}{\tilde{A}_t} \|x_{t-1} - x_t\| \|x_{t-1} - x_t\|
\end{aligned}
$$

Therefore,

$$
\boldsymbol{\alpha}\text{-REG}^y \leq 2L \sum_{t=1}^{T} \frac{\alpha_t^2}{\tilde{A}_t} \|x_{t-1} - x_t\|^2. \tag{22}
$$

For the x-player, its loss function in round $t$ is $\alpha_t \ell_t(x) := \alpha_t (\mu \phi(x) + \langle x, y_t \rangle)$, where $\phi(x) := \frac{1}{2} \|x\|_2^2$. Assume the x-player plays BETHEREGULARIZEDLEADER,

$$
x_t \leftarrow \arg\min_{x \in \mathcal{X}} \sum_{s=0}^{t} \alpha_s \ell_s(x), \tag{23}
$$

where $\alpha_0 \ell_0(x) := \alpha_0 \mu \phi(x)$. Denote

$$
\tilde{A}_t := \sum_{s=0}^{t} \alpha_s. \tag{24}
$$

Notice that this is different from $A_t := \sum_{s=1}^{t} \alpha_s$. Then, its regret is (proof is on the next page)

$$
\boldsymbol{\alpha}\text{-REG}^x := \sum_{t=1}^{T} \alpha_t \ell_t(x_t) - \alpha_t \ell_t(x^*) \leq \alpha_0 \mu L_0 \|x^* - x_0\| - \sum_{t=1}^{T} \frac{\mu \tilde{A}_{t-1}}{2} \|x_{t-1} - x_t\|^2, \tag{25}
$$

where $L_0$ is the Lipchitz constant of the 1-strongly convex function $\phi(x)$ and $x_0 = \arg\min_x \phi(x)$. Summing (22) and (25), we have

$$
\boldsymbol{\alpha}\text{-REG}^y + \boldsymbol{\alpha}\text{-REG}^x \leq \alpha_0 \mu L_0 \|x^* - x_0\| + \sum_{t=1}^{T} \left( \frac{2L\alpha_t^2}{A_t} - \frac{\mu \tilde{A}_{t-1}}{2} \right) \|x_{t-1} - x_t\|^2. \tag{26}
$$

We want to let the distance terms cancel out.

$$
\frac{2L\alpha_t^2}{\tilde{A}_t - a_0} - \frac{\mu \tilde{A}_{t-1}}{2} \leq 0, \tag{27}
$$

which is equivalent to

$$4L\alpha_t^2 \leq \mu \tilde{A}_t \tilde{A}_{t-1} - \mu\alpha_0 \tilde{A}_{t-1}.$$

$$4L\frac{\alpha_t^2}{\tilde{A}_t^2} \leq \mu\frac{\tilde{A}_{t-1}}{\tilde{A}_t} - \mu\alpha_0\frac{\tilde{A}_{t-1}}{\tilde{A}_t}\frac{1}{\tilde{A}_t} \tag{28}$$

$$4L\frac{\alpha_t^2}{\tilde{A}_t^2} \leq \mu(1 - \frac{\alpha_0}{\tilde{A}_t})(1 - \frac{\alpha_t}{\tilde{A}_t})$$

Let us denote the constant $\theta := \frac{\alpha_t}{\tilde{A}_t} > 0$.

$$\theta^2 + \frac{\mu}{4L}(1 - \frac{\alpha_0}{\tilde{A}_t})\theta - \frac{\mu}{4L}(1 - \frac{\alpha_0}{\tilde{A}_t}) \leq 0. \tag{29}$$

Notice that $0 < \frac{\alpha_0}{\tilde{A}_t} \leq 1$. It suffices to show that

$$\theta^2 + \frac{\mu}{4L}(1 - \frac{\alpha_0}{\tilde{A}_t})\theta - \frac{\mu}{4L} \leq 0. \tag{30}$$

Yet, we would expect that $\frac{\alpha_0}{\tilde{A}_t}$ is a decreasing function of $t$, so it suffices to show that

$$\theta^2 + \frac{\mu}{4L}(1 - \frac{\alpha_0}{\tilde{A}_1})\theta - \frac{\mu}{4L} \leq 0, \tag{31}$$

which is equivalent to

$$\theta^2 + \frac{\mu}{4L}\frac{\alpha_1}{\tilde{A}_1}\theta - \frac{\mu}{4L} \leq 0$$

$$\theta^2(1 + \frac{\mu}{4L}) - \frac{\mu}{4L} \leq 0. \tag{32}$$

It turns out that $\theta = \sqrt{\frac{\mu}{6L}} = \frac{1}{\sqrt{6\kappa}}$ satisfies the above inequality, combining the fact that $\frac{\mu}{L} \leq 1$.

Therefore, the optimization error $\epsilon$ after $T$ iterations is

$$\begin{aligned}
\epsilon &\leq \frac{\boldsymbol{\alpha}\text{-}\mathrm{REG}^y + \boldsymbol{\alpha}\text{-}\mathrm{REG}^x}{A_T} \leq \frac{1}{A_1}\frac{A_1}{A_2}\cdots\frac{A_{T-1}}{A_T}(\alpha_0\mu L_0\|x^* - x_0\|) \\
&= \frac{1}{A_1}(1 - \frac{\alpha_2}{A_2})\cdots(1 - \frac{\alpha_T}{A_T})(\alpha_0\mu L_0\|x^* - x_0\|) \\
&\leq \frac{1}{A_1}(1 - \frac{\alpha_2}{\tilde{A}_2})\cdots(1 - \frac{\alpha_T}{\tilde{A}_T})(\alpha_0\mu L_0\|x^* - x_0\|) \\
&\leq (1 - \frac{1}{\sqrt{6\kappa}})^{T-1}\frac{\alpha_0\mu L_0}{A_1}\|x^* - x_0\|.
\end{aligned} \tag{33}$$

which is $O((1 - \frac{1}{\sqrt{6\kappa}})^T) = O(\exp(-\frac{1}{\sqrt{6\kappa}}T))$.

$\square$

*Proof.* (of (25)) First, we are going to use induction to show that

$$\sum_{t=0}^{\tau} \alpha_t\ell_t(x_t) - \alpha_t\ell_t(x^*) \leq D_\tau, \tag{34}$$

for any $x^* \in \mathcal{X}$, where $D_\tau := -\sum_{t=1}^{\tau}\frac{\mu\tilde{A}_{t-1}}{2}\|x_{t-1} - x_t\|^2$.

For the base case $t = 0$, we have

$$\alpha_0\mu\phi(x_0) - \alpha_0\mu\phi(x^*) \leq 0 = D_0, \tag{35}$$

where $x_0$ is defined as $x_0 = \arg\min_{x\in\mathcal{X}} \alpha_0\mu\phi(x)$.

Now suppose it holds at $t = \tau - 1$.

$$\sum_{t=0}^{\tau} \alpha_t \ell_t(x_t) \overset{(a)}{\leq} D_{\tau-1} + \alpha_\tau \ell_\tau(x_\tau) + \sum_{t=0}^{\tau-1} \alpha_t \ell_t(x_{\tau-1})$$

$$\overset{(b)}{\leq} D_{\tau-1} + \alpha_\tau \ell_\tau(x_\tau) + \sum_{t=0}^{\tau-1} \alpha_t \ell_t(x_\tau) - \frac{\tilde{A}_{\tau-1}\mu}{2}\|x_{\tau-1} - x_\tau\|^2$$

$$= D_{\tau-1} + \sum_{t=0}^{\tau} \alpha_t \ell_t(x_\tau) - \frac{\tilde{A}_{\tau-1}\mu}{2}\|x_{\tau-1} - x_\tau\|^2 \qquad (36)$$

$$= D_\tau + \sum_{t=0}^{\tau} \alpha_t \ell_t(x_\tau)$$

$$\leq D_\tau + \sum_{t=0}^{\tau} \alpha_t \ell_t(x^*),$$

for any $x^* \in \mathcal{X}$, where $(a)$ we use the induction and we let the point $x^* = x_{\tau-1}$ and $(b)$ is by the strongly convexity and that $x_{\tau-1} = \arg\min_x \sum_{t=0}^{\tau-1} \alpha_t \ell_t(x)$ so that $\sum_{t=0}^{\tau-1} \alpha_t \ell_t(x_{\tau-1}) \leq \sum_{t=0}^{\tau-1} \alpha_t \ell_t(x_\tau) - \frac{\tilde{A}_{\tau-1}\mu}{2}\|x_{\tau-1} - x_\tau\|^2$ as $\sum_{t=0}^{\tau-1} \alpha_t \ell_t(x)$ is at least $\frac{\tilde{A}_{\tau-1}\mu}{2}$-strongly convex. We have completed the proof of (34). By (34), we have

$$\boldsymbol{\alpha}\text{-}\mathrm{REG}^x := \sum_{t=1}^{T} \alpha_t \ell_t(x_t) - \alpha_t \ell_t(x^*) \leq \alpha_0 \mu \phi(x^*) - \alpha_0 \mu \phi(x_0) - \sum_{t=1}^{T} \frac{\mu \tilde{A}_{t-1}}{2}\|x_{t-1} - x_t\|^2.$$

$$\leq \alpha_0 \mu L_0 \|x_0 - x^*\| - \sum_{t=1}^{T} \frac{\mu \tilde{A}_{t-1}}{2}\|x_{t-1} - x_t\|^2,$$

$$(37)$$

where we assume that $\phi(\cdot)$ is $L_0$-Lipchitz.

$\square$

# I Analysis of Accelerated Proximal Method

First, we need a stronger result.

**Lemma** [Property 1 in [6]] *For any proper lower semi-continuous convex function $\theta(x)$, let $x^+ = \mathrm{argmin}_{x \in \mathcal{K}} \theta(x) + V_c(x)$. Then, it satisfies that for any $x^* \in \mathcal{K}$,*

$$\theta(x^+) - \theta(x^*) \leq V_c(x^*) - V_{x^+}(x^*) - V_c(x^+). \qquad (38)$$

*Proof.* The statement and its proof has also appeared in [2] and [4]. For completeness, we replicate the proof here. Recall that the Bregman divergence with respect to the distance generating function $\phi(\cdot)$ at a point $c$ is: $V_c(x) := \phi(x) - \langle \nabla\phi(c), x - c \rangle - \phi(c)$.

Denote $F(x) := \theta(x) + V_c(x)$. Since $x^+$ is the optimal point of $\mathrm{argmin}_{x \in K} F(x)$, by optimality,

$$\langle x^* - x^+, \nabla F(x^+) \rangle = \langle x^* - x^+, \partial\theta(x^+) + \nabla\phi(x^+) - \nabla\phi(c) \rangle \geq 0, \qquad (39)$$

for any $x^* \in K$.

Now using the definition of subgradient, we also have

$$\theta(x^*) \geq \theta(x^+) + \langle \partial\theta(x^+), x^* - x^+ \rangle. \qquad (40)$$

By combining (39) and (40), we have

$$\theta(x^*) \geq \theta(x^+) + \langle \partial\theta(x^+), x^* - x^+ \rangle.$$

$$\geq \theta(x^+) + \langle x^* - x^+, \nabla\phi(c) - \nabla\phi(x^+) \rangle.$$

$$= \theta(x^+) - \{\phi(x^*) - \langle \nabla\phi(c), x^* - c \rangle - \phi(c)\} + \{\phi(x^*) - \langle \nabla\phi(x^+), x^* - x^+ \rangle - \phi(x^+)\}$$

$$+ \{\phi(x^+) - \langle \nabla\phi(c), x^+ - c \rangle - \phi(c)\}$$

$$= \theta(x^+) - V_c(x^*) + V_{x^+}(x^*) + V_c(x^+)$$

$$(41)$$

$\square$

Recall MirrorDescent 's update $x_t = \text{argmin}_x \gamma_t(\alpha_t h_t(x)) + V_{x_{t-1}}(x)$, where $h_t(x) = \langle x, y_t \rangle + \psi(x)$. Using the lemma with $\theta(x) = \gamma_t(\alpha_t h_t(x))$, $x^+ = x_t$ and $c = x_{t-1}$ we have that

$$\gamma_t(\alpha_t h_t(x_t)) - \gamma_t(\alpha_t h_t(x^*)) = \theta(x_t) - \theta(x^*) \leq V_{x_{t-1}}(x^*) - V_{x_t}(x^*) - V_{x_{t-1}}(x_t). \quad (42)$$

Therefore, we have that

$$
\begin{aligned}
\boldsymbol{\alpha}\text{-Reg}^x &:= \sum_{t=1}^T \alpha_t h_t(x_t) - \min_{x \in \mathcal{X}} \sum_{t=1}^T \alpha_t h_t(x) \\
&\overset{(42)}{\leq} \sum_{t=1}^T \frac{1}{\gamma_t}\big(V_{x_{t-1}}(x^*) - V_{x_t}(x^*) - V_{x_{t-1}}(x_t)\big) \\
&= \frac{1}{\gamma_1}V_{x_0}(x^*) - \frac{1}{\gamma_T}v_{x_T}(x^*) + \sum_{t=1}^{T-1}\big(\frac{1}{\gamma_{t+1}} - \frac{1}{\gamma_t}\big)V_{x_t}(x^*) - \frac{1}{\gamma_t}V_{x_{t-1}}(x_t) \\
&\overset{(a)}{\leq} \frac{1}{\gamma_1}D + \sum_{t=1}^{T-1}\big(\frac{1}{\gamma_{t+1}} - \frac{1}{\gamma_t}\big)D - \frac{1}{\gamma_t}V_{x_{t-1}}(x_t) = \frac{D}{\gamma_T} - \sum_{t=1}^T \frac{1}{\gamma_t}V_{x_{t-1}}(x_t) \\
&\overset{(b)}{\leq} \frac{D}{\gamma_T} - \sum_{t=1}^T \frac{1}{2\gamma_t}\|x_{t-1} - x_t\|^2,
\end{aligned}
\quad (43)
$$

where $(a)$ holds since the sequence $\{\gamma_t\}$ is non-increasing and $D$ upper bounds the divergence terms, and $(b)$ follows from the strong convexity of $\phi$, which grants $V_{x_{t-1}}(x_t) \geq \frac{1}{2}\|x_t - x_{t-1}\|^2$. Now we see that following the same lines as the proof in Section 3. We get that $\bar{x}_T$ is an $O(\frac{1}{T^2})$ approximate optimal solution.

## J   Accelerated FrankWolfe

---
**Algorithm 1** A new FW algorithm [[1]]

---
1: In the weighted loss setting of Algorithm 1:
2: **for** $t = 1, 2, \ldots, T$ **do**
3:     $y$-player uses OptimisitcFTL as OAlg$^x$: $y_t = \nabla f(\widetilde{x}_t)$.
4:     $x$-player uses BeTheRegularizedLeader with $R(X) := \frac{1}{2}\gamma_{\mathcal{K}}(x)^2$ as OAlg$^x$:
5:     Set $(\hat{x}_t, \rho_t) = \underset{x \in \mathcal{K}, \rho \in [0,1]}{\text{argmin}} \sum_{s=1}^t \rho\langle x, \alpha_s y_s \rangle + \frac{1}{\eta}\rho^2$ and play $x_t = \rho_t \hat{x}_t$.
6: **end for**

## K  Proof of Theorem 1

For completeness, we replicate the proof by [1] here.

**Theorem 1** *Assume a $T$-length sequence $\boldsymbol{\alpha}$ are given. Suppose in Algorithm 1 the online learning algorithms $OAlg^x$ and $OAlg^y$ have the $\boldsymbol{\alpha}$-weighted average regret $\overline{\boldsymbol{\alpha}\text{-}\mathrm{REG}}^x$ and $\overline{\boldsymbol{\alpha}\text{-}\mathrm{REG}}^y$ respectively. Then the output $(\bar{x}_T, \bar{y}_T)$ is an $\epsilon$-equilibrium for $g(\cdot, \cdot)$, with $\epsilon = \overline{\boldsymbol{\alpha}\text{-}\mathrm{REG}}^x + \overline{\boldsymbol{\alpha}\text{-}\mathrm{REG}}^y$.*

*Proof.* Suppose that the loss function of the $x$-player in round $t$ is $\alpha_t h_t(\cdot) : \mathcal{X} \to \mathbb{R}$, where $h_t(\cdot) := g(\cdot, y_t)$. The $y$-player, on the other hand, observes her own sequence of loss functions $\alpha_t \ell_t(\cdot) : \mathcal{Y} \to \mathbb{R}$, where $\ell_t(\cdot) := -g(x_t, \cdot)$.

$$
\begin{aligned}
\frac{1}{\sum_{s=1}^{T} \alpha_s} \sum_{t=1}^{T} \alpha_t g(x_t, y_t) &= \frac{1}{\sum_{s=1}^{T} \alpha_s} \sum_{t=1}^{T} -\alpha_t \ell_t(y_t) \\
&= -\frac{1}{\sum_{s=1}^{T} \alpha_s} \inf_{y \in \mathcal{Y}} \left\{ \sum_{t=1}^{T} \alpha_t \ell_t(y) \right\} - \frac{\boldsymbol{\alpha}\text{-}\mathrm{REG}^y}{\sum_{s=1}^{T} \alpha_s} \\
&= \sup_{y \in \mathcal{Y}} \left\{ \frac{1}{\sum_{s=1}^{T} \alpha_s} \sum_{t=1}^{T} \alpha_t g(x_t, y) \right\} - \overline{\boldsymbol{\alpha}\text{-}\mathrm{REG}}^y \\
\text{(Jensen)} \quad &\geq \sup_{y \in \mathcal{Y}} g\left( \tfrac{1}{\sum_{s=1}^{T} \alpha_s} \sum_{t=1}^{T} \alpha_t x_t, y \right) - \overline{\boldsymbol{\alpha}\text{-}\mathrm{REG}}^y \qquad (44) \\
&= \sup_{y \in \mathcal{Y}} g\left( \bar{x}_T, y \right) - \overline{\boldsymbol{\alpha}\text{-}\mathrm{REG}}^y \qquad (45) \\
&\geq \inf_{x \in \mathcal{X}} \sup_{y \in \mathcal{Y}} g\left( x, y \right) - \overline{\boldsymbol{\alpha}\text{-}\mathrm{REG}}^y
\end{aligned}
$$

Let us now apply the same argument on the right hand side, where we use the $x$-player's regret guarantee.

$$
\begin{aligned}
\frac{1}{\sum_{s=1}^{T} \alpha_s} \sum_{t=1}^{T} \alpha_t g(x_t, y_t) &= \frac{1}{\sum_{s=1}^{T} \alpha_s} \sum_{t=1}^{T} \alpha_t h_t(x_t) \\
&= \left\{ \sum_{t=1}^{T} \frac{1}{\sum_{s=1}^{T} \alpha_s} \alpha_t h_t(x) \right\} + \frac{\boldsymbol{\alpha}\text{-}\mathrm{REG}^x}{\sum_{s=1}^{T} \alpha_s} \\
&= \left\{ \sum_{t=1}^{T} \frac{1}{\sum_{s=1}^{T} \alpha_s} \alpha_t g(x^*, y_t) \right\} + \overline{\boldsymbol{\alpha}\text{-}\mathrm{REG}}^x \\
&\leq g\left( x^*, \sum_{t=1}^{T} \tfrac{1}{\sum_{s=1}^{T} \alpha_s} \alpha_t y_t \right) + \overline{\boldsymbol{\alpha}\text{-}\mathrm{REG}}^x \qquad (46) \\
&= g\left( x^*, \bar{y}_T \right) + \overline{\boldsymbol{\alpha}\text{-}\mathrm{REG}}^x \qquad (47) \\
&\leq \sup_{y \in \mathcal{Y}} g(x^*, y) + \overline{\boldsymbol{\alpha}\text{-}\mathrm{REG}}^x
\end{aligned}
$$

Note that $\sup_{y \in \mathcal{Y}} g(x^*, y) = f(x^*)$ be the definition of the game $g(\cdot, \cdot)$ and by Fenchel conjugacy, hence we can conclude that $\sup_{y \in \mathcal{Y}} g(x^*, y) = \inf_{x \in \mathcal{X}} \sup_{y \in \mathcal{Y}} g(x, y) = V^* = \sup_{y \in \mathcal{Y}} \inf_{x \in \mathcal{X}} g(x, y)$. Combining (45) and (47), we see that:

$$
\sup_{y \in \mathcal{Y}} g\left( \bar{x}_T, y \right) - \overline{\boldsymbol{\alpha}\text{-}\mathrm{REG}}^y \leq \inf_{x \in \mathcal{X}} g\left( x, \bar{y}_T \right) + \overline{\boldsymbol{\alpha}\text{-}\mathrm{REG}}^x
$$

which implies that $(\bar{x}_T, \bar{y}_T)$ is an $\epsilon = \overline{\boldsymbol{\alpha}\text{-}\mathrm{REG}}^x + \overline{\boldsymbol{\alpha}\text{-}\mathrm{REG}}^y$ equilibrium. $\qquad \square$

## Footnotes

[1] proposed a FrankWolfe like algorithm that not only requires a linear oracle but also enjoys $O(1/T^2)$ rate on all the known examples of strongly convex constraint sets that contain the origin, like $l_p$ ball and Schatten $p$ ball with $p \in (1, 2]$. Their analysis requires the assumption that the underlying function is also strongly-convex to get the fast rate. To describe their algorithm, denote $\mathcal{K}$ be any closed convex set that contains the origin. Define "gauge function" of $\mathcal{K}$ as $\gamma_{\mathcal{K}}(x) := \inf\{c \geq 0 : \frac{x}{c} \in \mathcal{K}\}$. Notice that, for a closed convex $\mathcal{K}$ that contains the origin, $\mathcal{K} = \{x \in \mathbb{R}^d : \gamma_{\mathcal{K}}(x) \leq 1\}$. Furthermore, the boundary points on $\mathcal{K}$ satisfy $\gamma_{\mathcal{K}}(x) = 1$.

[1] showed that the squared of a gauge function is strongly convex on the underlying $\mathcal{K}$ for all the known examples of strongly convex sets that contain the origin. Algorithm 1 is the algorithm. Clearly, Algorithm 1 is an instance of the meta-algorithm. We want to emphasize again that our analysis does not need the function $f(\cdot)$ to be strongly convex to show $O(1/T^2)$ rate. We've improved their analysis.