[Reviews · NeurIPS 2018]

Reviewer 1



The paper describes a reduction from minimizing smooth convex function to a Nash zero-sum game. It is shown that after T steps, the players are O(1/T^2)-close to an equilibrium. Then it is shown that the online to batch conversion here yields the well-known acceleration of GD due to Nesterov. What is new (and what was known)? 1. Known: the formulation of the game via Fenchel duality, where x-player plays MD and y-player plays optimisticFTL ([1]). 2. New: While [1] required the loss function to be both smooth and strongly convex, here only smoothness is required. This is a nice contribution. 3. New: showing that performing online-to-batch while using FTL rather than optimisticFTL, yields the HeavyBall algorithm. My only concern is that this view on acceleration is not new as several papers (mainly [2]) explored a similar primal-dual view. However, I still find some of the aspects (e.g., importance of being optimistic) interesting. Therefore, I lean towards accepting the paper. [1] Faster Rates for Convex-Concave Games [2] Linear Coupling: An Ultimate Unification of Gradient and Mirror Descent

Reviewer 2



This paper considered accelerating smooth convex function optimization via reduction to computing the Nash equilibrium of a particular zero-sum convex-concave game. The authors showed by using optimistic follow the leader, they can achieve a rate of convergence to Nash equilibrium to O(1/T^2), broadening the class of cost functions where this result holds compared to [1]. Furthermore, this result can be translated to the acceleration rate for minimizing smooth convex functions via Fenchel game. As a result, the authors showed that the algorithm derived in this framework with certain parameter setting coincide with the Nesterov acceleration algorithm. In addition, the authors showed that by modifying the parameters, they can derive several acceleration algorithms (e.g., HeavyBall, Nesterov’s 1-memory and \inf-memory), thus unifying them under the same framework. Overall, I enjoyed reading this paper, the presentation is clear, and I think it is interesting to the machine learning/optimization community to get a better understanding about the proof and intuition behind Nesterov’s acceleration results. This paper provides a neat and relatively simple framework to tie in a few widely recognized algorithms, which is beneficial for more researchers to understand the proofs of these algorithms.

Reviewer 3



This paper shows that Nesterov’s accelerated gradient descent algorithms can be interpreted as computing a saddle point via online optimization algorithms. A convex optimization problem is transformed to be a minmax problem by the Fenchel dual, the solution of which is then approximated via online optimization algorithms. This paper can be a significant contribution to the optimization community. I would say that this is one of the most natural interpretations of Nesterov’s accelerated gradient methods. The use of weighted regrets and (optimistic) FollowTheLeader (instead of follow the regularized leader) are a little bit artificial but acceptable. The latter is perhaps easier to accept, if the authors point out that \ell_t is strongly convex due to smoothness of f. A possible issue is boundedness of the radius D in Section 4. To use Theorem 2 and Corollary 1 to prove the accelerated convergence rate, one has to provide an upper bound of the radius D. It is not immediate to me whether D is actually bounded or not in Section 4. (In Section 4.2, there is a constraint set K, but boundedness of K is not mentioned.) Other comments: 1. ln 135: It seems that \ell_t(y) should be equal to -g(x_{t - 1}, y) instead of -g(x_t, y), according to Algorithm 1. Please check. 2. ln 4, Algorithm 2: The x in \nabal h_t (x) is not defined, though this is not really relevant. 3. Theorem 3: You need to specify \gamma to use Theorem 2 and Corollary 1.